# STRUCTURED TRANSFORMER CIRCUITS PRUNING

## ABSTRACT

Transformers become ubiquitous across vision and language tasks, but their depth and parameter count often far exceed what is needed for a given downstream application, leading to unnecessary compute and memory overhead. Existing layer-pruning techniques either require multiple retraining cycles, rely on continuous relaxations that never fully deactivate blocks, or depend on architecture-specific analyses. We introduce STCP, a model-agnostic, single-pass pruning framework that learns binary gates over each block's multi-head self-attention (MHSA) and MLP sub-layers in a pretrained transformer. We optimize gates while also injecting noise and introducing an $L_1$ penalty: this allows us to escape from local minima, and to find sparser circuits. We validate STCP on both image classification and NLP tasks with large pretrained models, showing good trade-offs in terms of complexity and performance. The code will be made publicly available upon acceptance of the article.

## 1 INTRODUCTION

Transformers have become the foundation of modern deep learning, driving breakthroughs in natural language processing, computer vision, and multimodal tasks. From language models like BERT (Kenton & Toutanova, 2019) and GPT-3 (Brown et al., 2020) to vision transformers for image recognition like CLIP (Radford et al., 2021) and ViT (Dosovitskiy et al., 2020), these architectures achieve state-of-the-art performance by leveraging multi-head self-attention (MHSA) and deep feed-forward networks across dozens or even hundreds of layers. However, this exceptional performance comes at the cost of enormous model sizes and computational demands (Strubell et al., 2020). The large number of parameters and layers in transformer models leads to high memory usage and latency, making deployment in resource-constrained environments difficult and raising concerns about energy consumption and carbon footprint.

With the growing concern about environmentally friendly AI, model compression techniques have gained increasing attention. Methods such as pruning (Lee et al., 2019; Tartaglione et al., 2022) and quantization (Jin et al., 2021) reduce the number or precision of parameters, while knowledge distillation (Wu et al., 2023) transfers knowledge from a large model to a smaller one. These approaches can significantly shrink model size or accelerate inference with minimal impact on accuracy.

However, most existing compression methods for transformers focus on reducing parameters or attention heads, rather than eliminating whole modules of the network (Tang et al., 2024). Removing only a handful of weights or heads often yields limited speed-ups, since the model's overall architecture and layer operations remain intact. Indeed, on modern hardware, computations within a transformer layer are highly parallelized, so the main bottleneck is the sequence of attention and feed-forward operations that each input must pass through in every layer. To substantially reduce inference time and resource usage, it is crucial to remove or bypass entire components in these layers (Ali Mehmeti-Göpel & Disselhoff, 2023). Meanwhile, naively pruning large components like an entire MHSA or MLP sub-layer can harshly degrade model performance (Sajjad et al., 2023). Moreover, prior approaches typically rely on iterative prune-and-finetune cycles or are designed for specific transformer components, which limits their practicality.

This paper addresses these challenges by introducing a novel method, Structured Transformer Circuits Pruning (STCP), which prunes a transformer model by jointly identifying and removing unimportant MHSA and MLP sub-layers after a single fine-tuning process, while preserving model accuracy.

STCP uses learnable gates on each MHSA and MLP sub-layer: during fine-tuning, these gates control the output of their modules. We add standard normal Gaussian noise to the gate activations to allow them to switch on and off, and apply $L_1$ regularization to encourage more gates to turn off over time. After fine-tuning, we sort all gate values and remove sub-layers in ascending order of gate value until the model's performance collapses. STCP showcases, among its strengths, that it does not require repeated retraining or fine-tuning after pruning and that it is agnostic to specific transformer architectures, making it applicable to a wide range of models and tasks. We empirically validate the effectiveness of STCP on several representative transformer models and benchmark tasks. Experimental results show that STCP can reduce models' complexity while maintaining their performance.

We summarize the key contributions and messages as follows.

- We present a binary gating mechanism for each transformer sub-layer, where gates combined with standard normal Gaussian noise and a straight-through gradient estimator learn which MHSA and MLP sub-layers to keep or remove in one fine-tuning pass (Sec. 3.1 and Sec. 3.2).
- We propose STCP, a model-agnostic pruning method that jointly optimizes gates, model weights, with an $L_1$ penalty on gates, allowing removal of redundant sub-layers without iterative retraining (Sec. 3.2 and Sec. 3.3).
- We demonstrate STCP on vision and language transformers across multiple benchmarks. Our results show that STCP can prune a significant number of sub-layers while preserving performance (Sec. 4).

## 2 RELATED WORKS

**Transformer circuits.** A neural circuit is a sparse computational subgraph within the model that captures specific aspects of its behaviour for analysis (Olah et al., 2020; Elhage et al., 2021). Circuit discovery aims to identify a sparse subgraph that explains how the full model behaves on a specific task (Olah et al., 2020). The transformer architecture is built from a series of MHSA and MLP layers, which operate on a shared residual stream (Elhage et al., 2021). This residual stream allows the model to remain functional even when some modules are removed. Understanding the internal circuits of transformers has motivated methods that prune model connections to isolate task-relevant sub-networks. Automated Circuit Discovery (ACDC) (Conmy et al., 2023), for example, identifies circuits by recursively pruning edges (connections) between edges that do not affect a chosen behavior. This recursive activation patching approach can successfully recover known mechanism circuits, but it incurs a high computational cost due to its iterative, edge-by-edge importance evaluations. EAP (Syed et al., 2023) is an attribution-based technique for circuit discovery: they compute the influence of each computational graph edge via attribution patching (a linearized gradient approximation of activation patching) and then remove the least important edges. This approach requires only one backward pass on the model, which achieves high efficiency at the cost of potentially suboptimal solutions to the circuit discovery problem. Edge Pruning (Bhaskar et al., 2024) is a method that leverages gradient-based signals to prune the edges between components (rather than entire neurons or layers) in one shot. The result is an extremely sparse subgraph that remains nearly as faithful to the full model's outputs as the original network. However, these interpretability-focused pruning methods operate at the level of individual connections within a fixed model. They aim to explain specific behaviors and generally do not produce a standalone compressed model.

**Structured pruning.** Beyond interpretability, a large body of work has explored structured pruning of transformers for model compression. In large language models (LLMs), researchers have pursued the removal of entire submodules or groups of parameters while preserving overall capabilities. LLM-Pruner (Ma et al., 2023) performs gradient-guided structural pruning on an LLM's weights, selectively removing non-critical coupled structures, and then uses a lightweight LoRA fine-tuning step to recover the model's performance. However, this work requires extra fine-tuning after pruning, and the entire MLP blocks cannot be removed. RECAP (Ilhan et al., 2024) is a method that iteratively prunes and fine-tunes a transformer in segments. In each cycle, RECAP prunes the model based on a Taylor approximation of weight importance and then updates only a subset of weights. This method introduces significant complexity and multiple retraining stages, and it can only remove channels rather than entire layers. OSSCAR (Meng et al., 2024) formulates one-shot structured pruning as a combinatorial optimization problem, minimizing layer-wise reconstruction error with local search

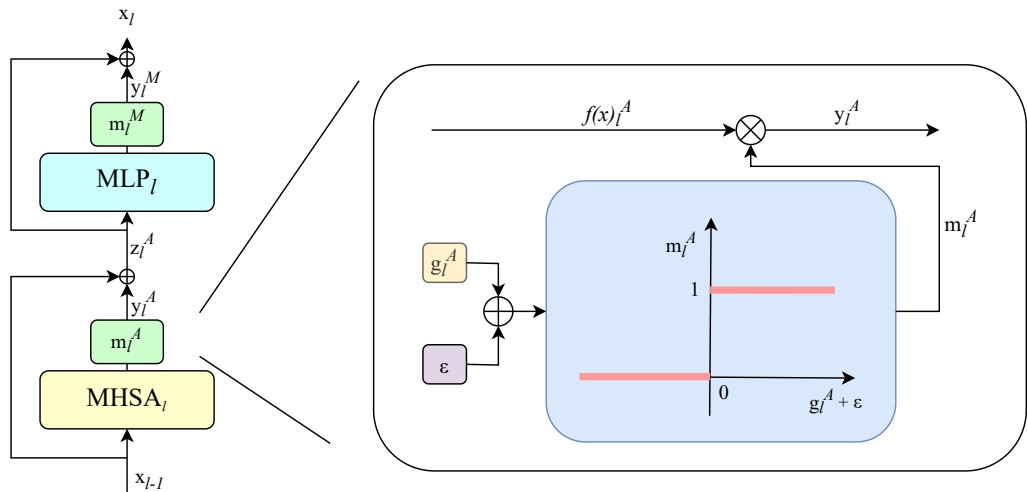

Figure 1: Forward pass for transformer encoder block with gates.

and low-rank updates to select sub-layer dimensions without fine-tuning. While OSSCAR shows scalability to large models, it is still not able to remove entire sub-layers. We will compare our method with the gradient-guided pruning and weight-guided pruning methods in Sec. 4.2.

**Layer removing.** Another line of research tries to remove entire layers or sub-layers from transformers when they are considered unnecessary. Recent methods (Lin et al., 2024; Zhang et al., 2024) usually identify MHSA sub-layers that contribute little information and then remove the MHSA sub-layer from selected blocks but retain and reuse parts of the MLP module. Some other approaches do aim to drop whole layers. For example, how pre-trained transformers respond to layer dropping without further training was investigated (Sajjad et al., 2023). They find that randomly removing layers during inference may have less impact than expected, especially in deeper models, suggesting redundancy in model depth. EASIER (Quétu et al., 2024) is a method that uses an entropy-based importance metric to rank entire layers in order of significance. They progressively remove the least important layers from an over-parameterized model and show that depth can be reduced to cut computation on easier tasks. TLC (Liao et al., 2025) uses greedy search to rank layers' importance and explore the boundary of the number of removable layers. Then, merge the least important layers with the following layers by removing nonlinearity. Shortened-Taylor (Kim et al., 2024) also uses greedy search to rank each module's importance and then removes the least important ones from the model. ShortGPT (Men et al., 2024) determines the importance of each module based on the differences in its inputs and outputs. Joint Layer Drop (He et al., 2024) is a fine-grained ShortGPT variant that applies the importance metric to prune attention and MLP sub-layers independently. These methods are effective but either require continuous relaxations of discrete pruning decisions or multi-stage retraining to gradually reach a compact model or facing challenge of maintaining performance after pruning. In contrast, STCP finds a pruned architecture in a single fine-tuning pass, no further training required after pruning, and models continue to perform well after pruning. We will also compare our method with the Shortened-Taylor and Joint Layer Drop in Sec. 4.2.

## 3 METHOD

In this section, we detail our method Structured Transformer Circuits Pruning (STCP), a model-agnostic, single-pass framework that learns to remove redundant multi-head self-attention (MHSA) and MLP sub-layers in pretrained transformers. We first introduce the binary gating mechanism (Sec. 3.1), then describe our learning strategy combining noise injection and $L_1$ regularization on gates (Sec. 3.2), and finally present the pruning procedure based on the learned gate values (Sec. 3.3). We conclude with an overview of the complete STCP algorithm (Sec. 3.4).

### 3.1 GATING SUB-LAYER OUTPUTS

Consider a transformer with $L$ encoder blocks, each containing a multi-head self-attention (MHSA) sub-layer and an MLP sub-layer. We denote a generic sub-layer by $f$ and its input by $x$. Without pruning, the forward pass is

$$y = f(x), \quad z = x + y. \tag{1}$$

To enable structured removal, we associate each sub-layer with a learnable scalar, we insert a learnable scalar $g$, which is activated by the Heaviside step function:

$$m = \mathbf{1}[g + \epsilon \geq 0], \tag{2}$$

where $\mathbf{1}[\cdot]$ is the Heaviside step and $\epsilon$ is standard normal Gaussian noise (see below). $g$ is initialized to a small positive value to preserve the network's behavior in the beginning, while ensuring each gate has a non-negligible chance to flip with the help of noise. Subsequently, we apply a gating mechanism to each sub-layer's outputs (as Fig. 1):

$$y = m \cdot f(x), \quad z = x + y. \tag{3}$$

Once $m$ equals 0, the corresponding sub-layer is removed from the forward pass.

Since the Heaviside step is non-differentiable and its true derivative is zero almost everywhere, naively backpropagating through the Heaviside step would cut off all gradient flow to the gate parameters. In backpropagation, we use the straight-through estimator to preserve a nonzero gradient path from the loss to $g$.

### 3.2 LEARNING GATES: NOISE INJECTION AND $\mathbf{L_1}$ REGULARIZATION

To enable each gate to explore both on and off states while still allowing gradient-based learning, and to escape from local minima, we inject standard normal Gaussian noise whose scale grows over a warm-up period. Then, for each gate we have

$$P(m = 1) = P(\epsilon > -g) = 1 - \Phi(-g), \tag{4}$$

Where $\Phi$ is the cumulative distribution function (CDF) of the standard normal. Thus, any gate with a small $|g|$ value has roughly a 50% chance to be on or off, enabling the network to discover which sublayers are truly dispensable. Equivalently, one can view the update as propagating through the expected mask:

$$\mathbb{E}[m] = 1 - \Phi(-g), \tag{5}$$

whose derivative with respect to $g$ is

$$\frac{\partial \mathbb{E}[m]}{\partial g} = \phi(-g), \tag{6}$$

where $\phi$ is the standard normal probability density function (PDF). This nonzero gradient path allows the optimizer to push each $g$ down if dropping the sub-layer does not harm the loss, or to hold it up if the sub-layer remains essential.

To bias the solution toward pruning, we add a small $L_1$ penalty on each gate parameter $g$ (or equivalently on the expected $m$ since $\mathbb{E}[m] \approx \text{Sigmoid}(g)$ in a continuous relaxation). This encourages gates to decrease towards 0. Denoting the set of all $2L$ gates by

$$\{g_1^A, \ldots, g_L^A, g_1^M, \ldots, g_L^M\},$$

we then optimize the joint cost function

$$\mathcal{L} = \underbrace{\mathcal{L}_{\text{task}}}_{\text{e.g. cross-entropy}} + \lambda \underbrace{\sum_{l=1}^{L} (|g_l^A| + |g_l^M|)}_{\text{L}_1 \text{ on gates}}, \tag{7}$$

where $\lambda > 0$ is constant throughout training.

---

**Algorithm 1** STCP: Learned Gating & Pruning

---
1: **Input:** pretrained model, data $\mathcal{D}$, $\lambda$, threshold.
2: Insert gates $g$ for all sub-layers.
3: **for** step $t = 1$ **to** $T$ **do**
4:     Compute masks $m = \mathbf{1}[g + \epsilon \geq 0]$.
5:     Forward/backward using $\mathcal{L}$ in Eq. (7), with straight-through estimator.
6:     Update gates and model's weights.
7: **end for**
8: Collect and sort the gate list in ascending.
9: **for** $k = 1$ **to** $2L$ **do**
10:     Permanently remove a sub-layer corresponding to the ranking.
11:     Evaluate on validation set.
12:     **if** accuracy drops below threshold **then break**
13:     **end if**
14: **end for**
15: **return** pruned model

---

### 3.3 SINGLE-PASS PRUNING VIA GATE RANKING

After the training of the gates is completed, each gate parameter $g$ has converged to a scalar value. Lower $g$ values indicate sub-layers that the model has learned to remove first. By ranking on learned gate magnitudes, we inherit the regularization and noise exploration of Sec. 3.2. Gates with large positive $g$ resist removal, preserving critical computation, while small or negative $g$ are dropped early. We prune sub-layers one at a time—removing the module with the smallest gate, checking validation accuracy immediately afterward, and stopping as soon as accuracy falls below the chosen threshold $\delta$.

This procedure yields a compact model in a single pass, avoiding repeated retraining. This strategy is architecture-agnostic and requires no specialized analysis of MHSA vs. MLP sub-layers.

### 3.4 OVERVIEW ON STCP

In Alg. 1, we present our method STCP that ranks transformer sub-layers' redundancy in one training pass, and then prunes sub-layers with minor (or even without) harm on the model's performance.

In our method, we first prepare the model with gate parameters (line 1- 2). Then run a single-pass fine-tuning: discrete masks $m$ are sampled (line 4) and the combined loss (line 5) trains both gates and weights (line 6).

After training, we sort all gate values (line 8) and remove sub-layers in order of increasing gate score (line 10), monitoring validation accuracy (line 11). We stop once the model can no longer tolerate further removals (line 12). This yields a compact transformer for fast inference and fewer computational costs, all without iterative retraining.

## 4 EXPERIMENTS

In this section, we empirically evaluate our proposed approach, STCP, across multiple architectures and datasets for image classification and natural language processing (NLP) setups.

### 4.1 EXPERIMENTAL SETUP

We validate our method through image classification and NLP tasks. Concerning image classification, our evaluation encompasses two models: CLIP (Radford et al., 2021) and ViT-H14 (Dosovitskiy et al., 2020). Models are trained on CIFAR-10 (Krizhevsky et al., 2009), Tiny-ImageNet (Le & Yang, 2015), ImageNet dataset (Deng et al., 2009), Flowers-102 (Nilsback & Zisserman, 2008), DTD (Cimpoi et al., 2014), Aircraft (Maji et al., 2013), and VLCS from DomainBed (Gulrajani & Lopez-Paz, 2020). For NLP, we apply our method on two models: BERT (Kenton & Toutanova, 2019) and RoBERTa (Liu et al., 2019). Models are trained on SST-2 (Socher et al., 2013) and QNLI (Williams et al., 2018).

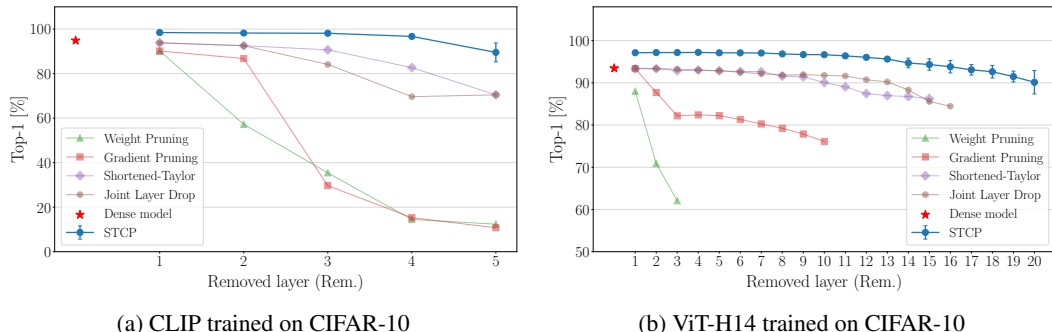

(a) CLIP trained on CIFAR-10        (b) ViT-H14 trained on CIFAR-10

Figure 2: Test performance (top-1) for models trained on CIFAR-10 with different numbers of layers removed by different methods.

We adopt the training setup from Wortsman et al. (2022). Unless stated otherwise, we use the default AdamW settings in PyTorch: $\beta_1 = 0.9$, $\beta_2 = 0.999$, and $\epsilon = 10^{-8}$. We employ different learning rates for different components during training: $\eta_{\text{gates}}, \eta_{\text{head}} > \eta_{\text{backbone}}$.

- **High rate for gates:** The gate parameters $g$ receive small gradient signals and can still move away quickly from zero to saturate into on/off regimes.
- **High rate for head:** As gates modify the feature flow, the classifier head needs flexibility to adapt to the evolving representation.
- **Low rate for backbone:** We preserve the fine-tuned pretrained weights, preventing destabilization from the noisy gating process.

The learning rate for the gates is set to $10^{-2}$, for the classification head to $10^{-3}$, and for the backbone to $10^{-6}$. We apply weight decay of 0.1 to the backbone and head, along with a cosine learning rate schedule from Loshchilov & Hutter (2016) that includes 500 warm-up steps. The length of the warm-up steps of noise is also set to 500. All gates are initially set to 0.5, and we use $\lambda = 10^{-6}$ for regularization. We tune the batch size to balance gradient variance and GPU memory, each specific setup corresponds to a different batch size. We also use gradient clipping with a global norm of 1, and each model is trained for 10,000 steps. All the trainings are performed on an NVIDIA A40 GPU equipped with 48GB RAM. All the experiments were repeated with four different random seeds, and the reported results include error bars (standard deviation) across these runs. The code is attached in the supplementary material and will be publicly available upon acceptance of the article. All the training details and parameter choices are provided in Appendix A.

We compare our results with the dense model and two baselines: Weight-Based Pruning and Gradient-Based Pruning, which remove sub-layers with the lowest weights/gradients. We also compare our method with two SOTA methods, Shortened-Taylor (Kim et al., 2024) and Joint Layer Drop (He et al., 2024). Moreover, for vision tasks, we compare our method with LaCoOT (Quétu et al., 2025). To make a fair comparison, for all baselines, we started from a pretrained model, and no fine-tuning was implemented after pruning.

## 4.2 RESULTS

**A first overview.** First, we tested our method STCP across different models trained on the CIFAR-10 dataset. Fig. 2 shows test accuracy for models trained on CIFAR-10 as we remove one sub-layer (MHSA or MLP) at each step with different methods. It appears that STCP closely matches or even slightly exceeds the dense model's accuracy for the early removals, and only begins to decline after several layers have been pruned, demonstrating high robustness. Shortened-Taylor and Joint Layer Drop

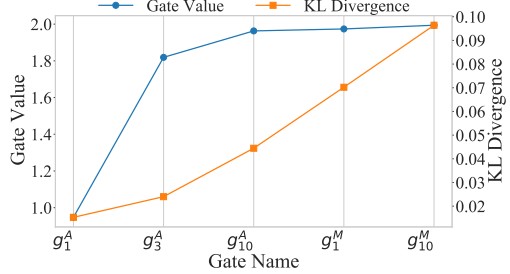

Figure 3: KL divergence between original and pruned CLIP_Flowers 102 outputs plotted against the gate values of the removed sub-layers.

Table 1: Test performance (Top-1) and the number of removed sub-layers (Rem.) for all image classification setups considered.

| Dataset | Approach | CLIP | | ViT-H14 | |
|---|---|---|---|---|---|
| | | Top-1 | Rem. | Top-1 | Rem. |
| CIFAR-10 | Dense model | 94.88 | 0/24 | 93.46 | 0/64 |
| | Weight Pruning | 57.25 | 2/24 | 70.88 | 2/64 |
| | Gradient Pruning | 90.17 | 1/24 | 82.18 | 3/64 |
| | Shortened-Taylor | 90.69 | 3/24 | 90.08 | 10/64 |
| | Joint Layer Drop | 84.12 | 3/24 | 91.62 | 10/64 |
| | LaCoOT | 85.44 | 2/24 | 92.22 | 8/64 |
| | STCP | **96.72 ± 0.51** | **4/24** | **93.82 ± 1.48** | **16/64** |
| Tiny-ImageNet | Dense model | 77.20 | 0/24 | 74.30 | 0/64 |
| | Weight Pruning | 67.32 | 1/24 | 64.44 | 1/64 |
| | Gradient Pruning | 67.32 | 1/24 | 64.44 | 1/64 |
| | Shortened-Taylor | 65.72 | 2/24 | 72.62 | 7/64 |
| | Joint Layer Drop | 60.7 | 2/24 | 69.78 | 7/64 |
| | LaCoOT | 69.86 | 1/24 | 72.64 | 8/64 |
| | STCP | **77.67 ± 5.32** | **3/24** | **74.48 ± 0.07** | **11/64** |
| VLCS | Dense model | 85.46 | 0/24 | 78.84 | 0/64 |
| | Weight Pruning | 84.06 | 1/24 | 67.57 | 2/64 |
| | Gradient Pruning | 82.95 | 1/24 | 72.79 | 6/64 |
| | Shortened-Taylor | 85.64 | 1/24 | 76.79 | 10/64 |
| | Joint Layer Drop | 83.41 | 1/24 | 77.17 | 10/64 |
| | LaCoOT | 67.19 | 1/24 | 76.05 | 11/64 |
| | STCP | **88.00 ± 0.64** | **2/24** | **78.29 ± 0.95** | **18/64** |
| DTD | Dense model | 76.12 | 0/24 | 70.11 | 0/64 |
| | Weight Pruning | 72.71 | 1/24 | 63.99 | 1/64 |
| | Gradient Pruning | 72.71 | 1/24 | 64.47 | 4/64 |
| | Shortened-Taylor | 70.64 | 2/24 | 69.57 | 7/64 |
| | Joint Layer Drop | 65.69 | 2/24 | 68.72 | 7/64 |
| | LaCoOT | 75.69 | 1/24 | 68.35 | 5/64 |
| | STCP | **78.96 ± 0.17** | **3/24** | **70.23 ± 0.46** | **11/64** |
| Flowers 102 | Dense model | 95.51 | 0/24 | 98.13 | 0/64 |
| | Weight Pruning | 91.14 | 1/24 | 94.83 | 1/64 |
| | Gradient Pruning | 91.14 | 1/24 | 86.11 | 3/64 |
| | Shortened-Taylor | 89.35 | 2/24 | 96.58 | 6/64 |
| | Joint Layer Drop | 84.29 | 2/24 | 96.62 | 7/64 |
| | LaCoOT | 76.53 | 1/24 | 95.12 | 5/64 |
| | STCP | **93.15 ± 2.24** | **2/24** | **98.22 ± 0.39** | **9/64** |
| Aircraft | Dense model | 52.87 | 0/24 | 47.85 | 0/64 |
| | Weight Pruning | 45.90 | 1/24 | 40.98 | 1/64 |
| | Gradient Pruning | 45.90 | 1/24 | 47.70 | 1/64 |
| | Shortened-Taylor | 45.60 | 1/24 | 47.67 | 7/64 |
| | Joint Layer Drop | 29.94 | 1/24 | 40.65 | 7/64 |
| | LaCoOT | 16.86 | 1/24 | 54.82 | 9/64 |
| | STCP | **60.81 ± 0.17** | **2/24** | **55.71 ± 0.32** | **17/64** |
| ImageNet | Dense model | 78.14 | 0/24 | 72.01 | 0/64 |
| | Weight Pruning | 74.31 | 1/24 | 65.40 | 1/64 |
| | Gradient Pruning | 74.31 | 1/24 | 65.40 | 1/64 |
| | Shortened-Taylor | 75.09 | 1/24 | 71.47 | 1/64 |
| | Joint Layer Drop | 69.42 | 1/24 | 71.51 | 1/64 |
| | LaCoOT | 43.15 | 1/24 | 71.91 | 2/64 |
| | STCP | **78.42 ± 5.10** | **2/24** | **72.14 ± 1.85** | **2/64** |

maintain strong accuracy initially, but drop more than STCP as pruning continues. Both gradient-based and weight-based pruning suffer obvious accuracy losses early on. In summary, STCP offers the best performance retention, followed by Shortened-Taylor and Joint Layer Drop, while gradient-based and weight-based methods degrade rapidly under aggressive pruning. The gates' values and

Table 2: Test performance (Top-1) and the number of removed layers (Rem.) for NLP setups.

| Dataset | Approach | BERT | | RoBERTa | |
|---|---|---|---|---|---|
| | | Top-1 | Rem. | Top-1 | Rem. |
| QNLI | Dense model | 90.61 | 0/24 | 91.47 | 0/24 |
| | Weight Pruning | 89.91 | 1/24 | 90.74 | 3/24 |
| | Gradient Pruning | 87.61 | 2/24 | 91.38 | 3/24 |
| | Shortened-Taylor | 89.64 | 2/24 | 91.63 | 3/24 |
| | Joint Layer Drop | 83.95 | 2/24 | 86.69 | 2/24 |
| | STCP | **90.23 ± 0.56** | **2/24** | **92.41 ± 0.04** | **4/24** |
| SST-2 | Dense model | 92.55 | 0/24 | 94.04 | 0/24 |
| | Weight Pruning | 92.43 | 2/24 | 93.81 | 3/24 |
| | Gradient Pruning | 91.86 | 2/24 | 94.04 | 2/24 |
| | Shortened-Taylor | 92.66 | 2/24 | 93.35 | 2/24 |
| | Joint Layer Drop | 92.20 | 2/24 | 92.55 | 2/24 |
| | STCP | **93.21 ± 0.17** | **2/24** | **94.18 ± 0.29** | **3/24** |

the possibility for each gate to flip after training for the CLIP model trained on the CIFAR-10 are presented in Appendix B.

For CLIP trained on Flowers 102 by our method, we tried to remove the sub-layers associated with the smallest gate values and measure the Kullback-Leibler (KL) divergence between the pruned and original outputs. In Fig. 3, we observe a clear, monotonic increase in KL divergence as the removed gate magnitudes grow. This trend indicates that sub-layers with lower gate values have minimal impact on the output. The consistent upward trend validates our gating mechanism's ability to identify and eliminate the least important components.

In the next paragraphs and sections, we show that across all image and language benchmarks, our STCP method consistently prunes a significant number of transformer sub-layers while keeping accuracy nearly unchanged.

**Image classification tasks.** Table 1 reports top-1 accuracy and number of removed layers (Rem.) for CLIP and ViT-H14 on seven image benchmarks and pruned by different methods. In each case, we also state the top-1 accuracy of the dense baseline (no pruning). It appears that STCP can prune up to 1/3 of the sub-layers and yet top-1 accuracy remains within a single point of the original. On easier datasets, it even edges accuracy slightly higher. For example, for CLIP model trained on CIFAR-10 dataset, our method can remove 4 sub-layers with around 2% top-1 accuracy increase, which indicates the existence of redundancy in the pre-trained model. On harder datasets, our method can still prune some sub-layers while keeping the models' functionality. Like in ImageNet-related tasks, both CLIP and ViT-H14 can remove 2 sub-layers while maintaining the models' performance without lowering. The result shows that STCP finds and removes redundant computation even when robustness is needed. It is also noticeable that in most cases, compared to other SOTA methods, STCP yields better results, either better top-1 accuracy, more removable layers, or both.

**NLP tasks.** Table 2 shows top-1 accuracy and removals for BERT and RoBERTa on two datasets. Similarly, in NLP tasks, STCP can effectively remove sub-layers and shows clear superiority over other SOTA methods. It appears that STCP can prune roughly $10\% - 20\%$ of the blocks in BERT and RoBERTa without more than a half-point accuracy drop. This demonstrates that, in one training pass, STCP reliably finds and removes redundant MHSA and MLP components in a wide range of models and datasets, producing leaner networks with negligible performance cost.

## 4.3 ABLATION STUDY

Table 3 provides an ablation study on the three key components identifiable within STCP: the designed gate mechanism in Sec. (3.1), the presence of standard normal Gaussian noise and the effect of regularization in Sec. (3.2). The result shows that every component contributes to the effectiveness of STCP.

Table 3: Ablation study on CLIP trained on CIFAR-10. Each component contributes to the effectiveness of STCP.

| Gate | Noise | Regularization | Top-1 | Rem. |
|------|-------|----------------|-------|------|
|      |       |                | 94.88 | 0/24 |
| ✓    |       |                | 91.93 | 3/24 |
| ✓    | ✓     |                | 97.43 | 3/24 |
| ✓    | ✓     | ✓              | 97.52 | 4/24 |

Table 4: Test performance (mIoU) and the number of removed layers (Rem.) for CLIP trained on Pascal VOC2012.

| Approach | mIoU  | Rem. |
|----------|-------|------|
| Dense    | 26.64 | 0/24 |
| STCP     | 26.89 | 1/24 |
| STCP     | 26.62 | 2/24 |
| STCP     | 19.05 | 3/24 |

Table 5: Top-1 and Rem. for ResNet-18 trained on CIFAR-10.

| Approach    | Top-1 | Rem. |
|-------------|-------|------|
| Dense model | 91.65 | 0/8  |
| STCP        | 91.47 | 1/8  |
| STCP        | 90.33 | 2/8  |
| STCP        | 86.83 | 3/8  |
| STCP        | 83.22 | 4/8  |

Table 6: GFLOPs, Inference time [ms], and Memory usage [MBs] of CLIP on Flowers 102 on a NVIDIA A4500.

| Rem. | GFLOPs | Inference time [ms] | Mem.usage [MB] | top-1 |
|------|--------|---------------------|----------------|-------|
| 0/24 | 36.62  | 8.87                | 583.89         | 95.51 |
| 1/24 | 35.44  | 8.59                | 574.69         | 96.98 |
| 2/24 | 33.58  | 8.56                | 556.68         | 95.33 |
| 3/24 | 32.41  | 8.10                | 547.67         | 91.72 |
| 4/24 | 30.55  | 8.04                | 529.65         | 89.90 |

We also apply STCP in segmentation. Table 4 reports mIoU and the number of removed sub-layers (Rem.) for CLIP trained on Pascal VOC2012 (Everingham et al., 2010). It appears that after 2 sub-layers are removed by STCP, the model can still maintain a good performance. This demonstrates that our method can be applied to more complex tasks. The gates' values and the possibility for each gate to flip after training for the CLIP model trained on the Pascal VOC2012 are presented in Appendix B.

To verify that STCP generalizes beyond transformers, we apply it to ResNet-18 (He et al., 2016) on CIFAR-10 by inserting gates at each residual block and pruning in ascending order of gate values. As reported in Table 5, removing one block only introduces a negligible drop (91.65% → 91.47%), while pruning two blocks still preserves 90.33% accuracy. We also trained Qwen2.5-0.5B on WikiText2 with STCP. As shown in Table 7, STCP can also effectively remove layers while maintaining model performance on modern-scale LLMs. The result demonstrates that our method is not limited to transformers, but also applicable to other models such as CNNs.

Table 6 showcases the potential savings in terms of FLOPS, inference time, and Memory usage on an NVIDIA A4500 GPU for a CLIP trained on Flowers 102 and pruned by STCP: the fewer layers the network has, the shorter the inference time, the smaller the number of FLOPs, and fewer the memory usage.

Quantization is also a popular model compression method. We believe my approach has the potential for further integration with quantization. To probe this, we applied post-training 16-bit and 8-bit quantization to our CLIP models on CIFAR-10 after STCP, and measured top-1 accuracy at different pruning levels. As shown in Table 8, the relative accuracy gap between the original full-precision model and 16/8-bit models remains small compared to the drop induced by pruning itself. It suggests that STCP-pruned models remain quantization-friendly, and these two model compression methods can be well combined.

## 4.4 LIMITATIONS

STCP attains competitive accuracy while pruning up to one-third of a transformer's modules. Meanwhile, some parameters like the gate initial value and the regularization weight $\lambda$ were picked by hand. Finding good, general rules for these parameters remains an open challenge. For example, He et al. (2023) employs grid search to determine some training parameters, but such sweeps are computationally expensive. We implemented ablation experiments for the hyperparameter choices in Appendix A. These experiments show that STCP is remarkably robust across a wide range of hyperparameter choices. While tuning these hyperparameters per architecture or dataset could introduce further

Table 7: Test performance (PPL) and the number of removed layers (Rem.) for Qwen2.5-0.5B trained on WikiText2.

| Approach | PPL | Rem. |
|---|---|---|
| Dense | 15.09 | 0/48 |
| STCP | 15.43 | 4/48 |
| STCP | 17.35 | 7/48 |
| STCP | 24.38 | 10/48 |
| STCP | 36.27 | 13/48 |
| STCP | 37.43 | 15/48 |

Table 8: Top-1 accuracy of STCP-pruned CLIP on CIFAR-10 for the original full-precision model (32 bit) and its 16-bit and 8-bit post-training quantized models.

| Rem. | 32-bit | 16-bit | 8-bit |
|---|---|---|---|
| Dense model | 94.88 | 94.88 | 94.80 |
| 1/24 | 98.40 | 98.40 | 98.32 |
| 2/24 | 98.12 | 98.12 | 98.00 |
| 3/24 | 98.02 | 98.02 | 97.87 |
| 4/24 | 97.52 | 97.52 | 97.24 |
| 5/24 | 84.63 | 84.61 | 83.61 |

gains, we used the same hyperparameter combination in all reported runs in our paper to conserve computational resources and simplify comparisons. An interesting next step could be to develop an automated criterion or heuristic for selecting the hyperparameters, which would eliminate the need for exhaustive grid searches and make STCP more effective in practice.

## 5 CONCLUSION

In this work, we introduced STCP, a method designed to efficiently prune transformer models by simultaneously targeting their MHSA and MLP sub-layers. By employing a learnable gating mechanism with standard normal Gaussian noise, STCP can identify and eliminate less important sub-layers in a single fine-tuning run while preserving high model performance. Our experiments on multiple transformer architectures and benchmarks demonstrate the robustness of STCP.

In our work, STCP lowers energy use and hardware cost, bringing large models to edge devices and greener data centers. We hope that in the future we can integrate STCP with other compression strategies to further push the boundaries of transformer efficiency.

## REPRODUCIBILITY STATEMENT

We provide all the training details and parameter choices in Sec. 4.1 and Appendix A. We have also cited the sources for all models and datasets used. Our source code is submitted as supplementary material and will be publicly available upon acceptance of the article.

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

## A EXPERIMENTAL DETAILS

The implementation details used in the experiments are presented here.

**Data augmentation.** CIFAR-10 is augmented with per-channel normalization, random horizontal flipping, and random shifting by up to four pixels in any direction. VLCS is augmented with per-channel normalization, random horizontal flipping, random cropping, and resizing to 224. The brightness, contrast, saturation, and hue are also randomly affected with a factor fixed to 0.4. Tiny ImageNet is augmented with per-channel normalization and random horizontal flipping. ImageNet is augmented with per-channel normalization, random horizontal flipping, random cropping, and resizing to 224. Flowers 102 is augmented with per-channel normalization, random horizontal and vertical flipping combined with a random rotation, and cropped to 224. DTD and Aircraft are augmented with random horizontal and vertical flipping, and with per-channel normalization. The sequence length of SST-2, QNLI, and RTE is set to 128.

**Training hyperparameters.** The training hyperparameters used in the experiments are presented in Table 9.

| Model | Dataset | Total steps | Batch | Optimizer | LR (gate/head/backbone) | Weight Decay | Scheduler | Warm-up steps | Clip Norm |
|-------|---------|-------------|-------|-----------|--------------------------|--------------|-----------|---------------|-----------|
| CLIP | CIFAR-10 | 10 000 | 128 | AdamW | 1e-2 / 1e-3 / 1e-6 | 0.1 | Cosine | 500 | 1.0 |
| CLIP | Tiny-ImageNet | 10 000 | 128 | AdamW | 1e-2 / 1e-3 / 1e-6 | 0.1 | Cosine | 500 | 1.0 |
| CLIP | ImageNet | 10 000 | 128 | AdamW | 1e-2 / 1e-3 / 1e-6 | 0.1 | Cosine | 500 | 1.0 |
| CLIP | Flowers-102 | 10 000 | 32 | AdamW | 1e-2 / 1e-3 / 1e-6 | 0.1 | Cosine | 500 | 1.0 |
| CLIP | DTD | 10 000 | 32 | AdamW | 1e-2 / 1e-3 / 1e-6 | 0.1 | Cosine | 500 | 1.0 |
| CLIP | Aircraft | 10 000 | 32 | AdamW | 1e-2 / 1e-3 / 1e-6 | 0.1 | Cosine | 500 | 1.0 |
| CLIP | VLCS | 10 000 | 16 | AdamW | 1e-2 / 1e-3 / 1e-6 | 0.1 | Cosine | 500 | 1.0 |
| ViT-H14 | CIFAR-10 | 10 000 | 32 | AdamW | 1e-2 / 1e-3 / 1e-6 | 0.1 | Cosine | 500 | 1.0 |
| ViT-H14 | Tiny-ImageNet | 10 000 | 32 | AdamW | 1e-2 / 1e-3 / 1e-6 | 0.1 | Cosine | 500 | 1.0 |
| ViT-H14 | ImageNet | 10 000 | 32 | AdamW | 1e-2 / 1e-3 / 1e-6 | 0.1 | Cosine | 500 | 1.0 |
| CLIP | Flowers-102 | 10 000 | 32 | AdamW | 1e-2 / 1e-3 / 1e-6 | 0.1 | Cosine | 500 | 1.0 |
| CLIP | DTD | 10 000 | 32 | AdamW | 1e-2 / 1e-3 / 1e-6 | 0.1 | Cosine | 500 | 1.0 |
| CLIP | Aircraft | 10 000 | 32 | AdamW | 1e-2 / 1e-3 / 1e-6 | 0.1 | Cosine | 500 | 1.0 |
| ViT-H14 | VLCS | 10 000 | 16 | AdamW | 1e-2 / 1e-3 / 1e-6 | 0.1 | Cosine | 500 | 1.0 |
| BERT | SST-2 | 10 000 | 32 | AdamW | 1e-2 / 1e-3 / 1e-6 | 0.1 | Cosine | 500 | 1.0 |
| BERT | QNLI | 10 000 | 32 | AdamW | 1e-2 / 1e-3 / 1e-6 | 0.1 | Cosine | 500 | 1.0 |
| RoBERTa | SST-2 | 10 000 | 32 | AdamW | 1e-2 / 1e-3 / 1e-6 | 0.1 | Cosine | 500 | 1.0 |
| RoBERTa | QNLI | 10 000 | 32 | AdamW | 1e-2 / 1e-3 / 1e-6 | 0.1 | Cosine | 500 | 1.0 |

Table 9: The different employed learning strategies.

**Parameters choice.** We implemented ablation experiments for the noise magnitude, initial gate value, and regularization strength $\lambda$ choices, as shown in Table 10, 11, and 12. These experiments show that STCP is remarkably robust across a wide range of hyperparameter choices.

Table 10: Test performance (Top-1) for CLIP trained on CIFAR10 and pruned by STCP with different noise magnitude.

| Noise magnitude | Top-1 |
|-----------------|-------|
| 0.5 | 93.23 |
| 1 | 97.52 |
| 2 | 96.70 |
| 5 | 97.46 |
| 10 | 93.70 |

Table 11: Test performance (Top-1) for CLIP trained on CIFAR10 and pruned by STCP with different initial gate values.

| Initial Gate Value | Top-1 |
|--------------------|-------|
| 0.1 | 97.46 |
| 0.3 | 97.33 |
| 0.5 | 97.52 |
| 1 | 97.99 |
| 2 | 96.11 |

## B MORE DETAILED RESULTS

**Gate convergence.** In Table 13 and 14, we provide here the gates' values and the possibility for each gate to flip after training for the CLIP model trained on the CIFAR10 and Pascal VOC2012 by STCP. Gates with stable small values have a high possibility of being in the inactive state (like the gate of 1st MHSA), removing them would not affect the model's performance much. While high-value gates have a very low possibility of being in an inactive state, removing them would

Table 12: Test performance (Top-1) for CLIP trained on CIFAR10 and pruned by STCP with different regularization strength $\lambda$.

| $\lambda$ | Top-1 |
|---|---|
| $1.0 \times 10^{-7}$ | 96.13 |
| $5.0 \times 10^{-7}$ | 96.08 |
| $1.0 \times 10^{-6}$ | 97.52 |
| $5.0 \times 10^{-6}$ | 96.08 |
| $1.0 \times 10^{-5}$ | 97.21 |
| $5.0 \times 10^{-5}$ | 97.41 |
| $1.0 \times 10^{-4}$ | 97.45 |
| $5.0 \times 10^{-4}$ | 96.82 |
| $1.0 \times 10^{-3}$ | 97.23 |
| $5.0 \times 10^{-3}$ | 97.75 |
| $1.0 \times 10^{-2}$ | 97.83 |
| $5.0 \times 10^{-2}$ | 95.50 |

largely destroy the model's performance. These results clearly show that after STCP training, some gates have negative values and some others also have a high probability of being in the inactive state, like the gate of 11th MLP for CLIP trained on CIFAR-10 and the gate of 0th MHSA for CLIP trained on Pascal VOC2012. At the same time, many gates escape from the area near 0, minimizing the probability of flipping, which highlights the importance of the sub-layers corresponding to these gates.

**Training time.** Table 15 presents the training cost for vanilla training, LaCoOT, and STCP in version setups. The results show that STCP increases memory and time requirements during training compared to vanilla training. However, since the parameters added by the gate are very small relative to the model's overall parameters, the increase in requirements is minimal.

Table 16 presents the single-pass training time for all the NLP setups. Our method implements pruning after this single-pass training, and no need for iterative retraining.

**Detailed experimental results.** Table 17, 18, 19, 20, 21, 22, 23, 24, 25, 26, 27, 28, 29, 30, 31, 32, 33, and 34 present the STCP result of different setups with each seed and the removed layer in each removing iteration. The tables show that STCP reliably flags low-impact sub-layers in a model- and task-specific way: for example, CLIP's block 11 MLP and block 1 MHSA, and ViT-H14's block 31 MHSA are often removed first with negligible accuracy loss. However, the "unimportant" sub-layers are different between different models.

## THE USE OF LARGE LANGUAGE MODELS

LLMs were used to polish writing and improve readability.

Table 13: Gates' value and inactive gate occurrence for CLIP after training on CIFAR10 by STCP.

| Sub-layer | Gate Value | Flip possibility |
|---|---|---|
| (0, MHSA) | 2.44 | 0.74% |
| (0, MLP) | 1.79 | 3.68% |
| (1, MHSA) | 0.01 | 49.78% |
| (1, MLP) | 1.98 | 2.41% |
| (2, MHSA) | 2.62 | 0.45% |
| (2, MLP) | 1.99 | 2.35% |
| (3, MHSA) | 2.74 | 0.31% |
| (3, MLP) | 2.27 | 1.16% |
| (4, MHSA) | 2.50 | 0.62% |
| (4, MLP) | 2.34 | 0.97% |
| (5, MHSA) | 2.57 | 0.51% |
| (5, MLP) | 2.61 | 0.46% |
| (6, MHSA) | 2.45 | 0.72% |
| (6, MLP) | 2.29 | 1.10% |
| (7, MHSA) | 2.02 | 2.19% |
| (7, MLP) | 2.33 | 1.00% |
| (8, MHSA) | 2.38 | 0.87% |
| (8, MLP) | 2.85 | 0.22% |
| (9, MHSA) | 2.41 | 0.79% |
| (9, MLP) | 2.96 | 0.16% |
| (10, MHSA) | 0.78 | 21.91% |
| (10, MLP) | 2.48 | 0.66% |
| (11, MHSA) | 3.23 | 0.06% |
| (11, MLP) | -0.01 | 50.25% |

Table 14: Gates' value and inactive gate occurrence for CLIP after training on Pascal VOC2012 by STCP.

| Sub-layer | Gate Value | Flip possibility |
|---|---|---|
| (0, MHSA) | -0.37 | 64.43% |
| (0, MLP) | 1.68 | 4.66% |
| (1, MHSA) | 0.04 | 48.28% |
| (1, MLP) | 9.97 | 0.00% |
| (2, MHSA) | 9.97 | 0.00% |
| (2, MLP) | 9.99 | 0.00% |
| (3, MHSA) | 1.94 | 2.61% |
| (3, MLP) | 9.88 | 0.00% |
| (4, MHSA) | 2.35 | 0.94% |
| (4, MLP) | 9.98 | 0.00% |
| (5, MHSA) | 9.16 | 0.00% |
| (5, MLP) | 9.99 | 0.00% |
| (6, MHSA) | 2.26 | 1.19% |
| (6, MLP) | 10.00 | 0.00% |
| (7, MHSA) | 1.73 | 4.20% |
| (7, MLP) | 9.69 | 0.00% |
| (8, MHSA) | 1.74 | 4.06% |
| (8, MLP) | 9.93 | 0.00% |
| (9, MHSA) | 2.04 | 2.09% |
| (9, MLP) | 9.80 | 0.00% |
| (10, MHSA) | 2.09 | 1.82% |
| (10, MLP) | 9.98 | 0.00% |
| (11, MHSA) | 0.30 | 38.13% |
| (11, MLP) | 0.50 | 30.83% |

Table 15: The peak memory usage during training and training time for image classification setups.

| Dataset | Approach | CLIP | | ViT-H14 | |
|---|---|---|---|---|---|
| | | Peak memory [GB] | Time | Peak memory [GB] | Time |
| CIFAR-10 | Vanilla training | 22.7 | 3h52 | 32.64 | 11h48 |
| | LaCoOT | 27.47 | 4h26 | 46.83 | 13h30 |
| | STCP | 26.64 | 4h09 | 36.96 | 11h07 |
| Tiny-ImageNet | Vanilla training | 22.59 | 3h39 | 32.64 | 6h52 |
| | LaCoOT | 27.54 | 3h53 | 46.64 | 10h11 |
| | STCP | 27.97 | 3h18 | 36.9 | 7h47 |
| VLCS | Vanilla training | 4.67 | 2h10 | 20.12 | 6h20 |
| | LaCoOT | 6.93 | 2h41 | 30.68 | 6h31 |
| | STCP | 5.39 | 2h19 | 22.04 | 6h13 |
| DTD | Vanilla training | 8.28 | 2h40 | 32.64 | 10h39 |
| | LaCoOT | 9.88 | 2h56 | 25.21 | 7h44 |
| | STCP | 10.42 | 3h19 | 36.96 | 14h47 |
| Flowers 102 | Vanilla training | 7.25 | 5h51 | 32.64 | 21h09 |
| | LaCoOT | 9.94 | 6h03 | 46.6 | 25h35 |
| | STCP | 9.07 | 6h00 | 36.97 | 21h47 |
| Aircraft | Vanilla training | 7.48 | 1h57 | 32.64 | 13h12 |
| | LaCoOT | 6.92 | 2h47 | 38.51 | 15h48 |
| | STCP | 8.86 | 2h34 | 36.85 | 15h14 |
| ImageNet | Vanilla training | 22.71 | 4h47 | 32.7 | 12h56 |
| | LaCoOT | 27.67 | 5h03 | 37.52 | 13h58 |
| | STCP | 26.64 | 4h18 | 37.05 | 12h49 |

Table 16: The single-pass training time for all considered NLP setups.

| Model | QNLI | SST-2 |
|---|---|---|
| BERT | 0h19 | 0h18 |
| RoBERTa | 0h20 | 0h19 |

Table 17: For CLIP trained on CIFAR-10 and pruned by STCP, test performance (Top-1) and the removed sub-layer in each iteration with different seeds.

| Rem. | Seed 0 | | Seed 1 | | Seed 2 | | Seed 3 | |
|---|---|---|---|---|---|---|---|---|
| | Sub-layer | Top-1 | Sub-layer | Top-1 | Sub-layer | Top-1 | Sub-layer | Top-1 |
| 1/24 | (11, MLP) | 98.47 | (11, MLP) | 98.5 | (11, MLP) | 98.4 | (11, MLP) | 98.44 |
| 2/24 | (1, MHSA) | 98.23 | (1, MHSA) | 98.23 | (1, MHSA) | 98.12 | (1, MHSA) | 98.33 |
| 3/24 | (10, MHSA) | 98.2 | (10, MHSA) | 98.14 | (10, MHSA) | 98.02 | (10, MHSA) | 98.17 |
| 4/24 | (0, MHSA) | 96.1 | (2, MLP) | 96.63 | (0, MLP) | 97.52 | (2, MLP) | 96.63 |
| 5/24 | (0, MLP) | 94.31 | (1, MLP) | 86.1 | (1, MLP) | 84.63 | (0, MLP) | 93.12 |
| 6/24 | (3, MHSA) | 78.59 | (0, MLP) | 42.96 | (2, MLP) | 46.66 | (1, MLP) | 36.65 |

Table 18: For CLIP trained on Tiny-Inet and pruned by STCP, test performance (Top-1) and the removed sub-layer in each iteration with different seeds.

| Rem. | Seed 0 | | Seed 1 | | Seed 2 | | Seed 3 | |
|---|---|---|---|---|---|---|---|---|
| | Sub-layer | Top-1 | Sub-layer | Top-1 | Sub-layer | Top-1 | Sub-layer | Top-1 |
| 1/24 | (11, MLP) | 84.48 | (1, MHSA) | 83.2 | (1, MHSA) | 83.24 | (11, MLP) | 84.52 |
| 2/24 | (1, MHSA) | 82.72 | (11, MLP) | 82.9 | (11, MLP) | 82.86 | (1, MHSA) | 82.92 |
| 3/24 | (3, MHSA) | 68.46 | (10, MHSA) | 80.98 | (10, MHSA) | 80.7 | (10, MHSA) | 80.54 |
| 4/24 | (10, MHSA) | 66.04 | (0, MLP) | 70.52 | (1, MLP) | 70.82 | (1, MLP) | 71.76 |

Table 19: For CLIP trained on VLCS and pruned by STCP, test performance (Top-1) and the removed sub-layer in each iteration with different seeds.

| Rem. | Seed 0 | | Seed 1 | | Seed 2 | | Seed 3 | |
|---|---|---|---|---|---|---|---|---|
| | Sub-layer | Top-1 | Sub-layer | Top-1 | Sub-layer | Top-1 | Sub-layer | Top-1 |
| 1/24 | (11, MLP) | 89.38 | (11, MLP) | 88.35 | (11, MLP) | 88.35 | (11, MLP) | 90.21 |
| 2/24 | (1, MHSA) | 88.35 | (1, MHSA) | 86.95 | (1, MHSA) | 88.07 | (1, MHSA) | 88.63 |
| 3/24 | (1, MLP) | 82.29 | (2, MLP) | 83.32 | (1, MLP) | 82.29 | (1, MLP) | 82.57 |
| 4/24 | (0, MHSA) | 80.71 | (2, MHSA) | 78.75 | (3, MLP) | 54.05 | (3, MHSA) | 62.53 |

Table 20: For CLIP trained on DTD and pruned by STCP, test performance (Top-1) and the removed sub-layer in each iteration with different seeds.

| Rem. | Seed 0 | | Seed 1 | | Seed 2 | | Seed 3 | |
|---|---|---|---|---|---|---|---|---|
| | Sub-layer | Top-1 | Sub-layer | Top-1 | Sub-layer | Top-1 | Sub-layer | Top-1 |
| 1/24 | (1, MHSA) | 80.00 | (1, MHSA) | 79.63 | (1, MHSA) | 79.84 | (1, MHSA) | 80.05 |
| 2/24 | (11, MLP) | 80.27 | (10, MHSA) | 78.78 | (10, MHSA) | 79.15 | (10, MHSA) | 78.99 |
| 3/24 | (10, MHSA) | 79.20 | (11, MLP) | 78.78 | (11, MLP) | 79.04 | (11, MLP) | 78.83 |
| 4/24 | (5, MHSA) | 74.10 | (0, MLP) | 71.97 | (1, MLP) | 62.93 | (2, MLP) | 72.71 |

Table 21: For CLIP trained on Flowers 102 and pruned by STCP, test performance (Top-1) and the removed sub-layer in each iteration with different seeds.

| Rem. | Seed 0 | | Seed 1 | | Seed 2 | | Seed 3 | |
|---|---|---|---|---|---|---|---|---|
| | Sub-layer | Top-1 | Sub-layer | Top-1 | Sub-layer | Top-1 | Sub-layer | Top-1 |
| 1/24 | (1, MHSA) | 97.07 | (1, MHSA) | 97.06 | (1, MHSA) | 96.88 | (1, MHSA) | 96.98 |
| 2/24 | (2, MLP) | 93.97 | (3, MHSA) | 93.92 | (1, MLP) | 89.40 | (10, MHSA) | 95.33 |
| 3/24 | (1, MLP) | 51.42 | (10, MHSA) | 91.80 | (10, MHSA) | 87.43 | (2, MLP) | 91.72 |
| 4/24 | (3, MHSA) | 53.23 | (1, MLP) | 43.18 | (2, MLP) | 53.29 | (10, MLP) | 89.90 |

Table 22: For CLIP trained on Aircraft and pruned by STCP, test performance (Top-1) and the removed sub-layer in each iteration with different seeds.

| Rem. | Seed 0 | | Seed 1 | | Seed 2 | | Seed 3 | |
|---|---|---|---|---|---|---|---|---|
| | Sub-layer | Top-1 | Sub-layer | Top-1 | Sub-layer | Top-1 | Sub-layer | Top-1 |
| 1/24 | (1, MHSA) | 69.55 | (1, MHSA) | 69.64 | (1, MHSA) | 68.95 | (1, MHSA) | 69.22 |
| 2/24 | (10, MHSA) | 60.76 | (10, MHSA) | 60.76 | (10, MHSA) | 60.64 | (10, MHSA) | 61.09 |
| 3/24 | (1, MLP) | 37.29 | (2, MLP) | 51.79 | (1, MLP) | 33.45 | (2, MLP) | 50.26 |
| 4/24 | (0, MHSA) | 30.39 | (0, MHSA) | 40.05 | (0, MLP) | 3.87 | (0, MLP) | 22.35 |

Table 23: For CLIP trained on ImageNet and pruned by STCP, test performance (Top-1) and the removed sub-layer in each iteration with different seeds.

| Rem. | Seed 0 | | Seed 1 | | Seed 2 | | Seed 3 | |
|---|---|---|---|---|---|---|---|---|
| | Sub-layer | Top-1 | Sub-layer | Top-1 | Sub-layer | Top-1 | Sub-layer | Top-1 |
| 1/24 | (11, MLP) | 80.08 | (11, MLP) | 80.15 | (11, MLP) | 80.10 | (11, MLP) | 80.10 |
| 2/24 | (1, MHSA) | 78.68 | (1, MLP) | 66.22 | (10, MLP) | 75.74 | (1, MHSA) | 78.63 |
| 3/24 | (1, MLP) | 62.20 | (1, MHSA) | 63.26 | (1, MLP) | 61.89 | (1, MLP) | 63.87 |
| 4/24 | (10, MLP) | 57.53 | (10, MLP) | 58.03 | (1, MHSA) | 58.44 | (10, MHSA) | 58.87 |

Table 24: For ViT-H14 trained on CIFAR-10 and pruned by STCP, test performance (Top-1) and the removed sub-layer in each iteration with different seeds.

| Rem. | Seed 0 | | Seed 1 | | Seed 2 | | Seed 3 | |
|---|---|---|---|---|---|---|---|---|
| | Sub-layer | Top-1 | Sub-layer | Top-1 | Sub-layer | Top-1 | Sub-layer | Top-1 |
| 1/64 | (29, MHSA) | 97.12 | (29, MHSA) | 97.10 | (29, MHSA) | 97.11 | (31, MHSA) | 97.21 |
| 2/64 | (31, MHSA) | 97.14 | (31, MLP) | 97.25 | (5, MHSA) | 97.00 | (31, MLP) | 97.28 |
| 3/64 | (12, MLP) | 97.13 | (31, MHSA) | 97.26 | (31, MHSA) | 97.05 | (29, MHSA) | 97.23 |
| 4/64 | (25, MLP) | 97.21 | (25, MLP) | 97.26 | (24, MLP) | 97.09 | (25, MLP) | 97.28 |
| 5/64 | (31, MLP) | 97.18 | (30, MHSA) | 97.17 | (25, MLP) | 97.08 | (5, MHSA) | 97.01 |
| 6/64 | (24, MLP) | 97.21 | (24, MLP) | 97.16 | (31, MLP) | 96.98 | (12, MLP) | 97.04 |
| 7/64 | (5, MHSA) | 97.11 | (5, MHSA) | 96.97 | (12, MLP) | 97.10 | (24, MLP) | 97.08 |
| 8/64 | (30, MHSA) | 96.99 | (21, MHSA) | 96.96 | (11, MHSA) | 96.76 | (11, MHSA) | 96.68 |
| 9/64 | (28, MHSA) | 96.71 | (27, MLP) | 96.89 | (20, MHSA) | 96.44 | (27, MLP) | 96.73 |
| 10/64 | (12, MLP) | 96.62 | (12, MLP) | 96.97 | (30, MLP) | 96.45 | (30, MHSA) | 96.61 |
| 11/64 | (27, MLP) | 96.60 | (29, MLP) | 96.53 | (26, MLP) | 96.26 | (29, MLP) | 96.16 |
| 12/64 | (29, MLP) | 96.16 | (11, MHSA) | 96.27 | (28, MHSA) | 95.98 | (12, MHSA) | 95.73 |
| 13/64 | (30, MLP) | 95.79 | (30, MLP) | 95.63 | (30, MHSA) | 95.60 | (23, MLP) | 95.52 |
| 14/64 | (23, MLP) | 95.66 | (26, MLP) | 95.28 | (29, MLP) | 95.11 | (7, MHSA) | 92.73 |
| 15/64 | (26, MLP) | 95.29 | (28, MHSA) | 94.98 | (23, MLP) | 95.04 | (30, MLP) | 91.99 |
| 16/64 | (14, MHSA) | 94.89 | (12, MHSA) | 94.49 | (27, MLP) | 94.62 | (28, MLP) | 91.27 |
| 17/64 | (11, MHSA) | 93.43 | (14, MHSA) | 93.39 | (21, MHSA) | 94.46 | (15, MLP) | 91.04 |
| 18/64 | (21, MHSA) | 93.29 | (15, MLP) | 93.17 | (12, MHSA) | 93.97 | (20, MHSA) | 90.09 |
| 19/64 | (20, MHSA) | 92.61 | (11, MLP) | 92.84 | (7, MHSA) | 90.73 | (28, MHSA) | 89.75 |
| 20/64 | (14, MLP) | 92.46 | (20, MHSA) | 92.22 | (15, MLP) | 90.30 | (9, MHSA) | 85.56 |

Table 25: For ViT-H14 trained on Tiny-Inet and pruned by STCP, test performance (Top-1) and the removed sub-layer in each iteration with different seeds.

| Rem. | Seed 0 | | Seed 1 | | Seed 2 | | Seed 3 | |
|------|---------|-------|---------|-------|---------|-------|---------|-------|
| | Sub-layer | Top-1 | Sub-layer | Top-1 | Sub-layer | Top-1 | Sub-layer | Top-1 |
| 1/64 | (31, MHSA) | 81.74 | (31, MHSA) | 81.46 | (31, MHSA) | 81.68 | (31, MHSA) | 81.60 |
| 2/64 | (31, MLP) | 81.40 | (31, MLP) | 81.58 | (31, MLP) | 81.34 | (27, MLP) | 81.62 |
| 3/64 | (23, MLP) | 81.56 | (30, MLP) | 81.10 | (23, MLP) | 81.32 | (31, MLP) | 81.32 |
| 4/64 | (30, MLP) | 81.10 | (27, MLP) | 80.72 | (27, MLP) | 81.16 | (23, MLP) | 81.18 |
| 5/64 | (30, MHSA) | 80.74 | (29, MLP) | 79.88 | (30, MLP) | 80.70 | (30, MLP) | 80.86 |
| 6/64 | (27, MLP) | 80.24 | (23, MLP) | 79.44 | (25, MLP) | 80.70 | (26, MLP) | 80.22 |
| 7/64 | (25, MLP) | 79.78 | (25, MLP) | 79.18 | (29, MLP) | 79.30 | (30, MHSA) | 79.80 |
| 8/64 | (29, MLP) | 78.48 | (26, MLP) | 78.24 | (26, MLP) | 78.54 | (24, MLP) | 79.30 |
| 9/64 | (26, MLP) | 77.70 | (30, MHSA) | 77.94 | (30, MHSA) | 77.82 | (25, MLP) | 78.88 |
| 10/64 | (1, MLP) | 75.38 | (1, MLP) | 75.08 | (1, MLP) | 75.46 | (29, MLP) | 77.06 |
| 11/64 | (24, MLP) | 74.46 | (24, MLP) | 74.36 | (24, MLP) | 74.54 | (1, MLP) | 74.54 |
| 12/64 | (28, MLP) | 72.66 | (28, MLP) | 72.60 | (28, MLP) | 72.60 | (28, MLP) | 72.66 |
| 13/64 | (5, MHSA) | 71.42 | (14, MLP) | 72.40 | (14, MLP) | 72.30 | (22, MLP) | 71.78 |
| 14/64 | (14, MLP) | 70.96 | (22, MLP) | 71.76 | (22, MLP) | 71.66 | (14, MLP) | 71.32 |
| 15/64 | (22, MLP) | 70.04 | (15, MLP) | 70.62 | (5, MHSA) | 70.34 | (15, MLP) | 70.94 |
| 16/64 | (15, MLP) | 69.24 | (5, MHSA) | 69.26 | (15, MLP) | 69.42 | (5, MHSA) | 69.40 |
| 17/64 | (29, MHSA) | 68.02 | (29, MHSA) | 68.06 | (16, MHSA) | 68.54 | (16, MHSA) | 68.70 |
| 18/64 | (16, MHSA) | 66.94 | (7, MHSA) | 61.74 | (10, MHSA) | 67.56 | (29, MHSA) | 66.90 |
| 19/64 | (7, MHSA) | 60.28 | (16, MHSA) | 60.64 | (7, MHSA) | 59.24 | (10, MHSA) | 66.00 |
| 20/64 | (10, MHSA) | 58.40 | (10, MHSA) | 58.20 | (29, MHSA) | 58.32 | (7, MHSA) | 58.08 |

Table 26: For ViT-H14 trained on VLCS and pruned by STCP, test performance (Top-1) and the removed sub-layer in each iteration with different seeds.

| Rem. | Seed 0 | | Seed 1 | | Seed 2 | | Seed 3 | |
|------|---------|-------|---------|-------|---------|-------|---------|-------|
| | Sub-layer | Top-1 | Sub-layer | Top-1 | Sub-layer | Top-1 | Sub-layer | Top-1 |
| 1/64 | (31, MHSA) | 83.50 | (31, MHSA) | 83.04 | (31, MHSA) | 80.89 | (31, MLP) | 83.32 |
| 2/64 | (31, MLP) | 83.97 | (31, MLP) | 82.48 | (31, MLP) | 81.08 | (31, MHSA) | 83.04 |
| 3/64 | (25, MLP) | 83.78 | (25, MLP) | 82.20 | (12, MLP) | 81.83 | (25, MLP) | 84.25 |
| 4/64 | (30, MLP) | 82.57 | (30, MLP) | 81.45 | (30, MLP) | 80.89 | (27, MLP) | 81.73 |
| 5/64 | (27, MLP) | 83.32 | (30, MHSA) | 81.08 | (25, MLP) | 81.73 | (29, MLP) | 81.45 |
| 6/64 | (29, MLP) | 82.48 | (12, MLP) | 80.06 | (28, MLP) | 81.55 | (30, MLP) | 83.22 |
| 7/64 | (28, MLP) | 82.20 | (29, MLP) | 80.52 | (29, MLP) | 80.80 | (12, MLP) | 82.57 |
| 8/64 | (12, MLP) | 82.39 | (27, MLP) | 80.15 | (30, MHSA) | 80.71 | (28, MLP) | 81.83 |
| 9/64 | (30, MHSA) | 81.17 | (28, MLP) | 80.15 | (27, MLP) | 80.80 | (23, MLP) | 82.11 |
| 10/64 | (23, MLP) | 83.60 | (26, MLP) | 78.94 | (23, MLP) | 81.17 | (26, MLP) | 81.45 |
| 11/64 | (24, MLP) | 81.83 | (23, MLP) | 80.52 | (14, MLP) | 80.24 | (30, MHSA) | 81.17 |
| 12/64 | (15, MLP) | 80.52 | (9, MLP) | 79.03 | (15, MLP) | 79.03 | (15, MLP) | 81.17 |
| 13/64 | (26, MLP) | 81.83 | (29, MHSA) | 78.94 | (24, MLP) | 80.24 | (24, MLP) | 80.34 |
| 14/64 | (8, MLP) | 80.99 | (15, MLP) | 78.01 | (26, MLP) | 79.03 | (9, MLP) | 81.55 |
| 15/64 | (14, MLP) | 80.62 | (22, MLP) | 78.66 | (17, MHSA) | 79.59 | (29, MHSA) | 78.94 |
| 16/64 | (17, MHSA) | 81.27 | (24, MLP) | 78.66 | (8, MLP) | 79.31 | (13, MHSA) | 79.96 |
| 17/64 | (13, MHSA) | 80.06 | (13, MHSA) | 78.94 | (1, MLP) | 78.19 | (8, MLP) | 79.50 |
| 18/64 | (1, MLP) | 79.03 | (10, MLP) | 78.38 | (11, MHSA) | 76.70 | (17, MHSA) | 79.03 |
| 19/64 | (7, MHSA) | 78.47 | (11, MHSA) | 77.45 | (4, MLP) | 76.42 | (1, MLP) | 79.31 |
| 20/64 | (9, MLP) | 75.96 | (14, MLP) | 76.51 | (13, MHSA) | 75.77 | (10, MLP) | 77.17 |
| 21/64 | (29, MHSA) | 77.63 | (23, MHSA) | 75.77 | (29, MHSA) | 75.49 | (28, MHSA) | 76.98 |
| 22/64 | (11, MHSA) | 76.23 | (17, MHSA) | 77.91 | (21, MHSA) | 77.54 | (14, MLP) | 76.79 |
| 23/64 | (16, MHSA) | 74.74 | (1, MLP) | 74.46 | (10, MLP) | 75.02 | (22, MLP) | 75.58 |
| 24/64 | (10, MLP) | 74.28 | (8, MLP) | 73.35 | (18, MHSA) | 74.37 | (7, MHSA) | 75.12 |
| 25/64 | (21, MHSA) | 72.41 | (7, MHSA) | 75.12 | (7, MLP) | 75.68 | (27, MHSA) | 74.74 |

Table 27: For ViT-H14 trained on DTD and pruned by STCP, test performance (Top-1) and the removed sub-layer in each iteration with different seeds.

| Rem. | Seed 0 | | Seed 1 | | Seed 2 | | Seed 3 | |
|------|-----------|-------|-----------|-------|-----------|-------|-----------|-------|
|      | Sub-layer | Top-1 | Sub-layer | Top-1 | Sub-layer | Top-1 | Sub-layer | Top-1 |
| 1/64  | (31, MHSA) | 72.50 | (30, MLP)  | 72.39 | (30, MLP)  | 72.71 | (30, MLP)  | 72.18 |
| 2/64  | (30, MLP)  | 72.61 | (31, MHSA) | 72.23 | (28, MLP)  | 72.82 | (28, MLP)  | 72.50 |
| 3/64  | (28, MLP)  | 72.77 | (28, MLP)  | 72.34 | (31, MHSA) | 72.55 | (31, MHSA) | 72.66 |
| 4/64  | (14, MHSA) | 72.93 | (14, MHSA) | 72.45 | (14, MHSA) | 72.50 | (14, MHSA) | 72.82 |
| 5/64  | (29, MLP)  | 72.23 | (30, MHSA) | 72.39 | (29, MLP)  | 72.50 | (13, MLP)  | 72.50 |
| 6/64  | (13, MLP)  | 71.86 | (13, MHSA) | 72.34 | (30, MLP)  | 72.02 | (29, MLP)  | 71.86 |
| 7/64  | (30, MHSA) | 71.76 | (13, MLP)  | 71.86 | (13, MHSA) | 72.07 | (30, MHSA) | 71.65 |
| 8/64  | (13, MHSA) | 71.54 | (29, MLP)  | 71.49 | (13, MLP)  | 71.76 | (13, MHSA) | 71.17 |
| 9/64  | (16, MHSA) | 70.85 | (16, MHSA) | 71.06 | (16, MHSA) | 71.12 | (16, MHSA) | 71.01 |
| 10/64 | (8, MHSA)  | 70.21 | (8, MHSA)  | 70.05 | (20, MLP)  | 71.12 | (8, MHSA)  | 70.16 |
| 11/64 | (15, MHSA) | 69.63 | (15, MLP)  | 70.43 | (15, MHSA) | 70.85 | (15, MLP)  | 70.00 |
| 12/64 | (15, MLP)  | 69.63 | (15, MHSA) | 69.31 | (8, MHSA)  | 69.10 | (15, MHSA) | 69.36 |
| 13/64 | (7, MHSA)  | 65.32 | (20, MLP)  | 69.68 | (15, MLP)  | 69.73 | (7, MHSA)  | 66.44 |
| 14/64 | (12, MHSA) | 60.80 | (7, MHSA)  | 65.69 | (22, MLP)  | 68.56 | (25, MLP)  | 65.05 |
| 15/64 | (22, MLP)  | 60.90 | (25, MLP)  | 64.89 | (7, MHSA)  | 65.48 | (22, MLP)  | 65.11 |

Table 28: For ViT-H14 trained on Flowers 102 and pruned by STCP, test performance (Top-1) and the removed sub-layer in each iteration with different seeds.

| Rem. | Seed 0 | | Seed 1 | | Seed 2 | | Seed 3 | |
|------|-----------|-------|-----------|-------|-----------|-------|-----------|-------|
|      | Sub-layer | Top-1 | Sub-layer | Top-1 | Sub-layer | Top-1 | Sub-layer | Top-1 |
| 1/64  | (31, MHSA) | 98.89 | (31, MHSA) | 98.81 | (31, MHSA) | 98.85 | (31, MHSA) | 98.86 |
| 2/64  | (30, MLP)  | 98.86 | (30, MLP)  | 98.83 | (30, MLP)  | 98.83 | (30, MLP)  | 98.86 |
| 3/64  | (15, MHSA) | 98.85 | (22, MLP)  | 98.80 | (30, MHSA) | 98.94 | (28, MHSA) | 98.81 |
| 4/64  | (30, MHSA) | 98.93 | (30, MHSA) | 98.88 | (14, MHSA) | 98.80 | (15, MHSA) | 98.73 |
| 5/64  | (29, MLP)  | 98.80 | (15, MHSA) | 98.86 | (31, MLP)  | 98.89 | (29, MLP)  | 98.72 |
| 6/64  | (28, MHSA) | 98.78 | (26, MLP)  | 98.81 | (28, MHSA) | 98.81 | (31, MLP)  | 98.68 |
| 7/64  | (10, MHSA) | 98.75 | (29, MLP)  | 98.68 | (29, MLP)  | 98.62 | (14, MHSA) | 98.54 |
| 8/64  | (14, MHSA) | 98.52 | (28, MLP)  | 98.29 | (13, MHSA) | 98.60 | (30, MHSA) | 98.63 |
| 9/64  | (31, MLP)  | 98.46 | (27, MLP)  | 97.58 | (23, MLP)  | 98.57 | (27, MLP)  | 98.28 |
| 10/64 | (28, MLP)  | 98.05 | (31, MLP)  | 97.15 | (10, MHSA) | 98.37 | (28, MLP)  | 97.72 |
| 11/64 | (16, MHSA) | 97.84 | (10, MHSA) | 96.91 | (15, MHSA) | 98.21 | (23, MLP)  | 97.53 |
| 12/64 | (26, MLP)  | 97.33 | (14, MHSA) | 96.68 | (28, MLP)  | 97.82 | (13, MHSA) | 97.43 |
| 13/64 | (27, MLP)  | 96.47 | (23, MLP)  | 96.29 | (27, MLP)  | 97.12 | (22, MLP)  | 97.33 |
| 14/64 | (22, MLP)  | 96.34 | (13, MHSA) | 96.16 | (22, MLP)  | 96.96 | (8, MHSA)  | 96.99 |
| 15/64 | (7, MHSA)  | 95.22 | (28, MHSA) | 96.11 | (25, MLP)  | 96.70 | (26, MLP)  | 96.16 |

Table 29: For ViT-H14 trained on Aircraft and pruned by STCP, test performance (Top-1) and the removed sub-layer in each iteration with different seeds.

| Rem. | Seed 0 | | Seed 1 | | Seed 2 | | Seed 3 | |
|---|---|---|---|---|---|---|---|---|
| | Sub-layer | Top-1 | Sub-layer | Top-1 | Sub-layer | Top-1 | Sub-layer | Top-1 |
| 1/64 | (28, MLP) | 61.36 | (31, MHSA) | 61.66 | (28, MLP) | 61.51 | (28, MLP) | 61.30 |
| 2/64 | (27, MLP) | 61.36 | (29, MLP) | 61.66 | (31, MHSA) | 61.51 | (31, MHSA) | 61.30 |
| 3/64 | (31, MHSA) | 61.36 | (28, MLP) | 61.66 | (29, MLP) | 61.51 | (27, MLP) | 61.30 |
| 4/64 | (29, MLP) | 61.36 | (27, MLP) | 61.33 | (27, MLP) | 61.51 | (29, MLP) | 61.30 |
| 5/64 | (30, MLP) | 61.06 | (30, MLP) | 61.48 | (30, MLP) | 61.24 | (30, MLP) | 61.36 |
| 6/64 | (31, MLP) | 61.24 | (26, MLP) | 60.94 | (31, MLP) | 61.36 | (30, MHSA) | 61.00 |
| 7/64 | (26, MLP) | 61.03 | (30, MHSA) | 60.70 | (30, MHSA) | 61.75 | (26, MLP) | 60.97 |
| 8/64 | (30, MHSA) | 61.27 | (31, MLP) | 61.21 | (26, MLP) | 61.39 | (31, MLP) | 61.06 |
| 9/64 | (24, MLP) | 61.33 | (13, MLP) | 60.55 | (24, MLP) | 61.48 | (24, MLP) | 61.27 |
| 10/64 | (5, MHSA) | 59.71 | (24, MLP) | 61.09 | (13, MLP) | 61.39 | (13, MLP) | 61.03 |
| 11/64 | (13, MLP) | 59.14 | (25, MLP) | 61.00 | (5, MHSA) | 59.35 | (9, MLP) | 60.28 |
| 12/64 | (11, MHSA) | 58.30 | (5, MHSA) | 58.90 | (11, MHSA) | 58.66 | (5, MHSA) | 58.78 |
| 13/64 | (9, MLP) | 57.67 | (22, MLP) | 58.57 | (22, MLP) | 58.48 | (25, MLP) | 58.30 |
| 14/64 | (18, MLP) | 57.40 | (9, MLP) | 58.18 | (9, MLP) | 57.91 | (11, MHSA) | 57.76 |
| 15/64 | (25, MLP) | 57.34 | (12, MHSA) | 57.22 | (18, MLP) | 57.25 | (18, MLP) | 57.58 |
| 16/64 | (12, MHSA) | 55.99 | (11, MHSA) | 55.75 | (25, MLP) | 57.07 | (23, MLP) | 56.95 |
| 17/64 | (22, MLP) | 56.11 | (18, MLP) | 55.27 | (12, MHSA) | 55.90 | (12, MHSA) | 55.57 |
| 18/64 | (23, MLP) | 55.33 | (23, MLP) | 55.30 | (17, MLP) | 55.12 | (22, MLP) | 55.39 |
| 19/64 | (13, MHSA) | 54.10 | (13, MHSA) | 53.98 | (15, MLP) | 54.04 | (20, MLP) | 54.94 |
| 20/64 | (20, MLP) | 53.35 | (7, MHSA) | 46.80 | (13, MHSA) | 53.29 | (16, MLP) | 52.96 |
| 21/64 | (16, MLP) | 51.28 | (20, MLP) | 46.08 | (23, MLP) | 52.96 | (17, MLP) | 51.28 |
| 22/64 | (15, MLP) | 50.68 | (16, MLP) | 44.19 | (14, MLP) | 52.15 | (7, MHSA) | 43.86 |
| 23/64 | (14, MLP) | 50.44 | (15, MLP) | 43.38 | (20, MLP) | 50.89 | (15, MLP) | 42.96 |
| 24/64 | (7, MHSA) | 41.88 | (17, MLP) | 41.49 | (7, MHSA) | 43.50 | (14, MLP) | 41.61 |
| 25/64 | (19, MLP) | 40.20 | (14, MLP) | 39.96 | (16, MLP) | 40.26 | (13, MHSA) | 39.78 |

Table 30: For ViT-H14 trained on ImageNet and pruned by STCP, test performance (Top-1) and the removed sub-layer in each iteration with different seeds.

| Rem. | Seed 0 | | Seed 1 | | Seed 2 | | Seed 3 | |
|---|---|---|---|---|---|---|---|---|
| | Sub-layer | Top-1 | Sub-layer | Top-1 | Sub-layer | Top-1 | Sub-layer | Top-1 |
| 1/64 | (31, MLP) | 74.56 | (31, MLP) | 74.20 | (31, MLP) | 74.36 | (31, MLP) | 74.16 |
| 2/64 | (1, MLP) | 70.35 | (21, MHSA) | 73.92 | (21, MHSA) | 74.06 | (1, MLP) | 70.22 |
| 3/64 | (21, MHSA) | 69.96 | (1, MLP) | 69.82 | (1, MLP) | 69.82 | (21, MHSA) | 69.75 |
| 4/64 | (25, MHSA) | 69.02 | (26, MHSA) | 69.20 | (28, MHSA) | 69.21 | (28, MHSA) | 69.24 |
| 5/64 | (26, MHSA) | 67.87 | (23, MHSA) | 68.15 | (23, MHSA) | 68.41 | (26, MHSA) | 68.29 |
| 6/64 | (23, MHSA) | 66.47 | (24, MHSA) | 66.88 | (26, MHSA) | 67.33 | (24, MHSA) | 67.02 |
| 7/64 | (28, MHSA) | 64.88 | (25, MHSA) | 64.07 | (24, MHSA) | 65.52 | (25, MHSA) | 64.55 |
| 8/64 | (20, MHSA) | 63.47 | (28, MHSA) | 61.85 | (25, MHSA) | 61.99 | (20, MHSA) | 63.37 |
| 9/64 | (24, MHSA) | 60.02 | (20, MHSA) | 59.60 | (20, MHSA) | 59.86 | (23, MHSA) | 59.62 |
| 10/64 | (27, MHSA) | 54.76 | (23, MLP) | 59.32 | (22, MHSA) | 54.35 | (23, MLP) | 59.22 |

Table 31: For BERT trained on QNLI and pruned by STCP, test performance (Top-1) and the removed sub-layer in each iteration with different seeds.

| Rem. | Seed 0 | | Seed 1 | | Seed 2 | | Seed 3 | |
|---|---|---|---|---|---|---|---|---|
| | Sub-layer | Top-1 | Sub-layer | Top-1 | Sub-layer | Top-1 | Sub-layer | Top-1 |
| 1/24 | (2, MHSA) | 90.46 | (11, MLP) | 90.74 | (1, MHSA) | 90.59 | (11, MLP) | 90.55 |
| 2/24 | (1, MHSA) | 89.25 | (1, MHSA) | 90.48 | (11, MLP) | 90.61 | (2, MHSA) | 90.55 |
| 3/24 | (11, MLP) | 89.27 | (2, MHSA) | 89.18 | (0, MHSA) | 89.05 | (0, MHSA) | 90.41 |
| 4/24 | (3, MLP) | 82.48 | (3, MLP) | 83.03 | (3, MLP) | 88.36 | (1, MHSA) | 87.68 |
| 5/24 | (8, MLP) | 75.78 | (7, MHSA) | 80.08 | (2, MHSA) | 80.18 | (8, MHSA) | 86.44 |

Table 32: For BERT trained on SST-2 and pruned by STCP, test performance (Top-1) and the removed sub-layer in each iteration with different seeds.

| Rem. | Seed 0 | | Seed 1 | | Seed 2 | | Seed 3 | |
|------|-----------|-------|-----------|-------|-----------|-------|-----------|-------|
| | Sub-layer | Top-1 | Sub-layer | Top-1 | Sub-layer | Top-1 | Sub-layer | Top-1 |
| 1/24 | (1, MHSA) | 93.23 | (3, MHSA) | 93.12 | (1, MHSA) | 93.46 | (1, MHSA) | 93.23 |
| 2/24 | (3, MHSA) | 93.00 | (1, MHSA) | 93.12 | (3, MHSA) | 93.23 | (3, MHSA) | 93.46 |
| 3/24 | (5, MHSA) | 92.32 | (0, MHSA) | 91.97 | (0, MHSA) | 92.09 | (0, MHSA) | 91.97 |
| 4/24 | (0, MHSA) | 90.94 | (2, MHSA) | 88.88 | (2, MHSA) | 88.88 | (4, MHSA) | 90.83 |
| 5/24 | (2, MLP)  | 88.42 | (4, MHSA) | 84.75 | (5, MHSA) | 85.89 | (5, MHSA) | 87.50 |
| 6/24 | (2, MHSA) | 84.86 | (5, MHSA) | 80.96 | (1, MLP)  | 85.67 | (2, MHSA) | 80.50 |
| 7/24 | (4, MHSA) | 80.39 | (2, MLP)  | 80.39 | (4, MHSA) | 80.16 | (2, MLP)  | 79.93 |

Table 33: For RoBERTa trained on QNLI and pruned by STCP, test performance (Top-1) and the removed sub-layer in each iteration with different seeds.

| Rem. | Seed 0 | | Seed 1 | | Seed 2 | | Seed 3 | |
|------|------------|-------|------------|-------|------------|-------|------------|-------|
| | Sub-layer | Top-1 | Sub-layer | Top-1 | Sub-layer | Top-1 | Sub-layer | Top-1 |
| 1/24 | (10, MLP)  | 92.40 | (10, MLP)  | 92.40 | (10, MLP)  | 92.33 | (10, MLP)  | 92.29 |
| 2/24 | (11, MLP)  | 92.40 | (11, MLP)  | 92.42 | (11, MLP)  | 92.35 | (9, MLP)   | 92.35 |
| 3/24 | (9, MLP)   | 92.40 | (10, MHSA) | 92.44 | (9, MLP)   | 92.31 | (11, MLP)  | 92.39 |
| 4/24 | (10, MHSA) | 92.42 | (1, MHSA)  | 92.39 | (10, MHSA) | 92.37 | (10, MHSA) | 92.48 |
| 5/24 | (4, MHSA)  | 90.76 | (2, MHSA)  | 91.18 | (3, MHSA)  | 91.09 | (8, MLP)   | 92.22 |
| 6/24 | (3, MHSA)  | 87.59 | (9, MLP)   | 91.20 | (2, MHSA)  | 88.98 | (4, MHSA)  | 90.26 |
| 7/24 | (1, MHSA)  | 85.67 | (3, MHSA)  | 86.77 | (1, MHSA)  | 86.27 | (0, MHSA)  | 87.64 |

Table 34: For RoBERTa trained on SST-2 and pruned by STCP, test performance (Top-1) and the removed sub-layer in each iteration with different seeds.

| Rem. | Seed 0 | | Seed 1 | | Seed 2 | | Seed 3 | |
|------|-----------|-------|-----------|-------|-----------|-------|-----------|-------|
| | Sub-layer | Top-1 | Sub-layer | Top-1 | Sub-layer | Top-1 | Sub-layer | Top-1 |
| 1/24 | (11, MLP)  | 94.38 | (11, MLP) | 94.84 | (11, MLP) | 94.61 | (11, MLP) | 94.72 |
| 2/24 | (10, MLP)  | 94.38 | (10, MLP) | 94.72 | (10, MLP) | 94.61 | (10, MLP) | 94.61 |
| 3/24 | (5, MHSA)  | 94.61 | (2, MHSA) | 93.81 | (3, MHSA) | 94.15 | (5, MHSA) | 94.15 |
| 4/24 | (0, MHSA)  | 93.23 | (3, MHSA) | 93.35 | (5, MHSA) | 93.81 | (2, MHSA) | 93.58 |
| 5/24 | (4, MHSA)  | 92.20 | (5, MHSA) | 92.66 | (4, MHSA) | 92.66 | (3, MHSA) | 93.00 |
| 6/24 | (3, MHSA)  | 91.40 | (4, MHSA) | 92.09 | (8, MHSA) | 92.20 | (1, MHSA) | 91.40 |
| 7/24 | (2, MHSA)  | 88.53 | (9, MLP)  | 90.94 | (0, MHSA) | 89.79 | (4, MHSA) | 89.45 |

