# OpenReview forum: "Structured Transformer Circuits Pruning"
_ICLR.cc/2026/Conference — Submitted to ICLR 2026_

### Official Review · Reviewer_qoVD · 2025-10-27

**Soundness:** 3
**Presentation:** 2
**Contribution:** 3
**Rating:** 6
**Confidence:** 3

**Summary:**

This paper proposes Structured Transformer Circuits Pruning (STCP), a structured pruning method to improve the efficiency of Transformer models by removing entire Multi-Head Self-Attention (MHSA) and MLP sub-layers within each block. They apply a binary gate to each sub-layer to decide whether to turn it on or off. This gate is implemented using the non-differentiable Heaviside step function, and it employs the Straight-Through Estimator (STE) trick to allow gradients to flow during backpropagation. To help the gates effectively explore on and off states during fine-tuning, Gaussian noise is injected. Simultaneously, an L1 regularization penalty is applied to encourage the gate values to move toward zero. After training, gate parameters $g$ are ranked, and sub-layers are removed sequentially until performance drops just below a threshold. Experiments across various Vision (CLIP, ViT) and NLP (BERT, RoBERTa) tasks demonstrated that the method is highly robust in terms of its performance-to-compression ratio compared to existing methods, validating the effectiveness of the proposed methodology.

**Strengths:**

1. The method is well-motivated by applying a binary gating mechanism to structured pruning. This directly targets real-world computational and memory overheads, which is often not achieved by less-structured pruning.

2. STCP is a single-pass pruning framework. This makes it significantly simpler and more practical than prior approaches that typically rely on complex, multi-stage, iterative prune-and-finetune cycles.

3. The framework demonstrates broad applicability. It is model-agnostic and shows strong, consistent performance across diverse architectures (e.g., CLIP, ViT, BERT, ROBERTa) and different domains, including both Vision and NLP tasks.

4. The method proves to be highly robust. Extensive experiments show that it can prune a significant number of sub-layers while maintaining performance very close to, or in some cases even slightly exceeding, the original dense model.

**Weaknesses:**

1. The final pruning stage is a greedy approach that ranks gate values and removes sub-layers sequentially. This method does not guarantee finding the optimal combination of remaining sub-layers.
2. The process requires manual intervention. The user must define an acceptable performance drop threshold to determine when to stop removing layers, which is not an automated part of the framework.
3. The paper lacks sufficient references and comparisons to other established differentiable gating mechanisms used for pruning (e.g., Gumbel-Softmax).
4. The overall writing, structure, and presentation require further polish and refinement. This includes insufficient detail in the Method section (Sec 3), redundant sentences (e.g., “The gates’ values and the possibility for each gate to flip after training for the CLIP model trained on the CIFAR-10 are presented in Appendix B.” is duplicated on pages 7 and 8), and imprecise statements (e.g., “Thus, any gate with $|g|$ small has roughly a 50% chance to be on or off, …” on page 4).

**Questions:**

Refer to Weaknesses

---

> ### Author Response · Authors · 2025-11-19
>
> **[W1 - Gates learn during sing-pass]** Our “single-pass” formulation means that we learn all gate parameters jointly in one fine-tuning run, obtain a stable ranking of sub-layer importance, and prune once, with no post-pruning retraining required.
>
> Running a new optimization after every removed layer would require L sequential fine-tuning runs for L layers, which is computationally expensive for large transformer models. During training, the L1 regularizer reduces the gates’ magnitude and increases their flip probability under the Two-Sigmoid relaxation. This mechanism reveals a ranking of sub-layer importance, making additional retraining stages unnecessary.
> - - -
>
> **[W2 - Trade-off between performance and compression]** Our method automatically produces a trade-off curve between performance and compression as layers are removed in order of ranked importance. In practice, this curve allows users to select the operating point that matches their computational budget or acceptable performance drop.
> - - -
>
> **[W3 - More sufficient comparisons]** We further compare our approach with LaCoOT [R1], a novel method which is recently published at ICCV 2025. The results are shown in Table R1 and have been integrated into Table 1 of the revised paper. The results demonstrate that our method exhibits significant advantages over other established mechanisms in terms of the number of layers removed while maintaining model performance.
>
> *Table R1: Test performance (Top-1) and the number of removed sub-layers (Rem.) for all image classification tasks trained by LaCoOT and STCP.*
> | Dataset        | Approach | CLIP Top-1     | CLIP Rem. | ViT-H14 Top-1    | ViT-H14 Rem. |
> |----------------|----------|----------------|-----------|------------------|---------------|
> | CIFAR-10       | LaCoOT   | 85.44          | 2/24      | 92.22            | 8/64          |
> | CIFAR-10       | STCP     | 96.72±0.51     | 4/24      | 93.82±1.48       | 16/64         |
> | Tiny-ImageNet  | LaCoOT   | 69.86          | 1/24      | 72.64            | 8/64          |
> | Tiny-ImageNet  | STCP     | 77.67±5.32     | 3/24      | 74.48±0.07       | 11/64         |
> | VLCS           | LaCoOT   | 67.19          | 1/24      | 76.05            | 11/64         |
> | VLCS           | STCP     | 88.00±0.64     | 2/24      | 78.29±0.95       | 18/64         |
> | DTD            | LaCoOT   | 75.69          | 1/24      | 68.35            | 5/64          |
> | DTD            | STCP     | 78.96±0.17     | 3/24      | 70.23±0.46       | 11/64         |
> | Flowers 102    | LaCoOT   | 76.53          | 1/24      | 95.12            | 5/64          |
> | Flowers 102    | STCP     | 93.15±2.24     | 2/24      | 98.22±0.39       | 9/64          |
> | Aircraft       | LaCoOT   | 16.86          | 1/24      | 54.82            | 9/64          |
> | Aircraft       | STCP     | 60.81±0.17     | 2/24      | 55.71±0.32       | 17/64         |
> | ImageNet       | LaCoOT   | 43.15          | 1/24      | 71.91            | 2/64          |
> | ImageNet       | STCP     | 78.42±5.10     | 2/24      | 72.14±1.85       | 2/64          |
> - - -
>
> **[W4 - Improve writing]** Thank you for pointing them out. We improved them in the revised paper.
> - - -
>
> [R1]: Quétu, Victor, et al. "LaCoOT: Layer collapse through optimal transport." Proceedings of the IEEE/CVF International Conference on Computer Vision. 2025.

---

> > ### Comment · Reviewer_qoVD · 2025-11-27
> >
> > Thank you for your thoughtful response. While some of my concerns have not been fully addressed, the major issues have been sufficiently considered. Therefore, I will maintain my original score.

---

### Official Review · Reviewer_TxxT · 2025-10-30

**Soundness:** 3
**Presentation:** 3
**Contribution:** 2
**Rating:** 2
**Confidence:** 3

**Summary:**

This manuscript proposes and evaluates a procedure for removing entire components of transformer models. The scheme adds "gates" to the model and then jointly optimizes parameters of the gates and model weights to identify components that can be removed. Numerical experiments are provided to illustrate the efficacy of the scheme.

**Strengths:**

The manuscript provides an interesting method for shrinking the size of transformer models while retaining performance. Moreover, in the realm of methods that remove entire components it seems to outperform other methods when no fine-tuning is allowed.

**Weaknesses:**

The main weakness of this work is its narrowness in terms of how it presents the comparative landscape. While I agree with the manuscript that there are good computational reasons to preference removal of entire components within a model vs. unstructured pruning methods (in terms of practical speedup), it does seem that the competitive landscape is ultimately larger than just methods that can remove entire components. For example structured pruning methods (i.e., those that remove entire neurons) can be effective for MLPs since they retain the computational structure of the layer just at a smaller size (as opposed to, e.g., introducing unstructured sparsity).

The literature is too numerous to name, but, e.g., [Kwon, Woosuk, Sehoon Kim, Michael W. Mahoney, Joseph Hassoun, Kurt Keutzer, and Amir Gholami. "A fast post-training pruning framework for transformers." Advances in Neural Information Processing Systems 35 (2022): 24101-24116.] and [van der Ouderaa, Tycho FA, Markus Nagel, Mart Van Baalen, and Tijmen Blankevoort. "The LLM Surgeon." In The Twelfth International Conference on Learning Representations.]

One reason to bring this up is that a cursory comparison of results with Kwon et al. above and https://arxiv.org/pdf/2406.00061 suggests that, e.g., BERT models can have a substantial fraction ~40% of the parameters removed in a structured matter while retaining performance similar to the proposed method when only 2 layers are removed. (Thought it is hard to make an explicit comparison across papers because it seems the pretrained baseline may differ?)

In addition, quantization is also a very effective strategy for reducing both storage and computational time. While, presumably, a pruned model could then be quantized, it is not clear that these procedures are orthogonal: perhaps an aggressively pruned model resists quantization in a way that makes it better to quantize the larger model than prune then quantize. While it is unreasonable to burden this manuscript with answering that rather broad question, it does illustrate that a broader range of experiments are likely needed to help situate the proposed procedure in the broader landscape. I consider comparisons with structured pruning somewhat essential, quantization less so.

Lastly, while I am sympathetic to the ease with which one ask for "more experiments," I do think that newer, larger models (particularly for NLP tasks) should be considered to understand both the generality and scalability of the method. Perhaps (smaller) Llama models or similar would be useful here to provide at least a few points of comparison with other modern methods (e.g., LLM surgeon or similar)

Some assorted minor comments/weaknesses:

- While the proposed method does not have an explicit "fine tuning" step, the weights are updated with the gates (assuming the pseudo code is followed). (In fact, presumably this is why removing a layer can increase performance.) As such, it seems a bit unfair to completely omit fine tuning from other methods; better would be to control for compute and balance that (or, report compute to make it easier to assess the relative expense of methods).

**Questions:**

- While the method is articulated as "single-pass," presumably once a certain layer is removed that could change the optimal $g$ parameters for the other layers significantly, is there any value to thinking about the method via sweeps in which at each step 1 or a few layers are removed and then the gates are retrained?

- The $\ell_1$ penalty seemingly drives the $g$ towards 0, but that corresponds to at 50/50 gate (if I interpreted things correctly) and not an "off" gate, were other regularization strategies or penalties explored?

---

> ### Author Response · Authors · 2025-11-19
>
> **[W1 - Method Effectiveness]** We agree that our pruning ratio is smaller than the sparsity reported from the methods in the review mentioned above. However, these approaches prune at a much finer granularity (weights, channels, heads) inside each MHSA/MLP sub-layer and always keep the full transformer depth, whereas our method removes entire MHSA or MLP sub-layers, so the resulting network has strictly shorter forward paths.
>
> Prior works [R2, R3] on structured pruning show that coarse sparsity is more hardware-friendly than fine-grained sparsity. And in order to substantially reduce inference time and resource usage, it is crucial to remove or bypass entire components in these layers [R4]. In this sense, our method is complementary to fine-grained approaches: they maximise nominal sparsity at small granularity, while we push sparsity at the level of whole sub-layers in a way that is directly useful for deployment-time acceleration.
> - - -
>
> **[W2 - Combine STCP with quantization]** We thank the reviewer for raising the interaction between pruning and quantization. We also agree that a pruned model could then be quantized. To probe this, we applied post-training 16-bit and 8-bit quantization to our CLIP models on CIFAR-10 after STCP, and measured top-1 accuracy at different pruning levels.
> As shown in Table R5, the relative accuracy gap between the original full-precision model and 16/8- bit models remains small compared to the drop induced by pruning itself. It suggests that STCP-pruned models remain quantization-friendly, and these two model compression methods can be well combined. We have also included this result in Section 4.3 of the revised paper.
>
> *Table R5: Top-1 accuracy of STCP-pruned CLIP on CIFAR-10 for the original full-precision model (32-bit) and its 16-bit and 8-bit post-training quantized models.*
> | Rem.        | 32-bit | 16-bit | 8-bit |
> |-------------|--------|--------|--------|
> | Dense model | 94.88  | 94.88  | 94.80 |
> | 1/24        | 98.40  | 98.40  | 98.32 |
> | 2/24        | 98.12  | 98.12  | 98.00 |
> | 3/24        | 98.02  | 98.02  | 97.87 |
> | 4/24        | 97.52  | 97.52  | 97.24 |
> | 5/24        | 84.63  | 84.61  | 83.61 |
> - - -
>
> **[W3 - Generality and scalability on larger models]** We trained Qwen2.5-0.5B on WikiText2 with STCP. As shown in Table R3, STCP can also effectively remove layers while maintaining model performance on modern-scale LLMs. We have also included this result in Section 4.3 of the revised version.
>
> Moreover, we further compare our approach with LaCoOT [R1], a novel method which is recently published at ICCV 2025. The results are shown in Table R1 and have been integrated into Table 1 of the revised paper.  The results demonstrate that our method exhibits significant advantages over other SOTA approaches in terms of the number of layers removed while maintaining model performance.
>
> *Table R3: Test performance (PPL) and the number of removed layers (Rem.) for Qwen2.5-0.5B trained on WikiText2.*
> | Approach | PPL   | Rem.  |
> |----------|--------|--------|
> | Dense    | 15.09  | 0/48   |
> | STCP     | 15.43  | 4/48   |
> | STCP     | 17.35  | 7/48   |
> | STCP     | 24.38  | 10/48  |
> | STCP     | 36.27  | 13/48  |
> | STCP     | 37.43  | 15/48  |
>
> *Table R1: Test performance (Top-1) and the number of removed sub-layers (Rem.) for all image classification tasks trained by LaCoOT and STCP.*
> | Dataset        | Approach | CLIP Top-1     | CLIP Rem. | ViT-H14 Top-1    | ViT-H14 Rem. |
> |----------------|----------|----------------|-----------|------------------|---------------|
> | CIFAR-10       | LaCoOT   | 85.44          | 2/24      | 92.22            | 8/64          |
> | CIFAR-10       | STCP     | 96.72±0.51     | 4/24      | 93.82±1.48       | 16/64         |
> | Tiny-ImageNet  | LaCoOT   | 69.86          | 1/24      | 72.64            | 8/64          |
> | Tiny-ImageNet  | STCP     | 77.67±5.32     | 3/24      | 74.48±0.07       | 11/64         |
> | VLCS           | LaCoOT   | 67.19          | 1/24      | 76.05            | 11/64         |
> | VLCS           | STCP     | 88.00±0.64     | 2/24      | 78.29±0.95       | 18/64         |
> | DTD            | LaCoOT   | 75.69          | 1/24      | 68.35            | 5/64          |
> | DTD            | STCP     | 78.96±0.17     | 3/24      | 70.23±0.46       | 11/64         |
> | Flowers 102    | LaCoOT   | 76.53          | 1/24      | 95.12            | 5/64          |
> | Flowers 102    | STCP     | 93.15±2.24     | 2/24      | 98.22±0.39       | 9/64          |
> | Aircraft       | LaCoOT   | 16.86          | 1/24      | 54.82            | 9/64          |
> | Aircraft       | STCP     | 60.81±0.17     | 2/24      | 55.71±0.32       | 17/64         |
> | ImageNet       | LaCoOT   | 43.15          | 1/24      | 71.91            | 2/64          |
> | ImageNet       | STCP     | 78.42±5.10     | 2/24      | 72.14±1.85       | 2/64          |

---

> > ### Author Response · Authors · 2025-11-19
> >
> > **[W4 - Training cost]** In Table R2, we compare the training costs of vanilla training, LaCoOT, and STCP. The training cost has also been integrated into Table 15 of the revised paper. The results show that STCP increases memory and time requirements during training compared to vanilla training. However, since the number of parameters added by the gate is very small relative to the model's overall parameter amount, the increase in requirements is minimal. And STCP continuously requires less training cost than LaCoOT.
> >
> > *Table R2: The peak memory usage during training and training time for image classification setups.*
> > | Dataset        | Approach         | CLIP Peak Memory (GB) | CLIP Time | ViT-H14 Peak Memory (GB) | ViT-H14 Time |
> > |----------------|------------------|------------------------|-----------|---------------------------|---------------|
> > | CIFAR-10       | Vanilla training | 22.7                   | 3h52      | 32.64                    | 11h48        |
> > | CIFAR-10       | LaCoOT           | 27.47                  | 4h26      | 46.83                    | 13h30        |
> > | CIFAR-10       | STCP             | 26.64                  | 4h09      | 36.96                    | 11h07        |
> > | Tiny-ImageNet  | Vanilla training | 22.59                  | 3h39      | 32.64                    | 6h52         |
> > | Tiny-ImageNet  | LaCoOT           | 27.54                  | 3h53      | 46.64                    | 10h11        |
> > | Tiny-ImageNet  | STCP             | 27.97                  | 3h18      | 36.9                     | 7h47         |
> > | VLCS           | Vanilla training | 4.67                   | 2h10      | 20.12                    | 6h20         |
> > | VLCS           | LaCoOT           | 6.93                   | 2h41      | 30.68                    | 6h31         |
> > | VLCS           | STCP             | 5.39                   | 2h19      | 22.04                    | 6h13         |
> > | DTD            | Vanilla training | 8.28                   | 2h40      | 32.64                    | 10h39        |
> > | DTD            | LaCoOT           | 9.88                   | 2h56      | 25.21                    | 7h44         |
> > | DTD            | STCP             | 10.42                  | 3h19      | 36.96                    | 14h47        |
> > | Flowers 102    | Vanilla training | 7.25                   | 5h51      | 32.64                    | 21h09        |
> > | Flowers 102    | LaCoOT           | 9.94                   | 6h03      | 46.6                     | 25h35        |
> > | Flowers 102    | STCP             | 9.07                   | 6h00      | 36.97                    | 21h47        |
> > | Aircraft       | Vanilla training | 7.48                   | 1h57      | 32.64                    | 13h12        |
> > | Aircraft       | LaCoOT           | 6.92                   | 2h47      | 38.51                    | 15h48        |
> > | Aircraft       | STCP             | 8.86                   | 2h34      | 36.85                    | 15h14        |
> > | ImageNet       | Vanilla training | 22.71                  | 4h47      | 32.7                     | 12h56        |
> > | ImageNet       | LaCoOT           | 27.67                  | 5h03      | 37.52                    | 13h58        |
> > | ImageNet       | STCP             | 26.64                  | 4h18      | 37.05                    | 12h49        |
> > - - -
> >
> > **[Q1 & Q2 - Gates learn during sing-pass]** Our “single-pass” formulation means that we learn all gate parameters jointly in one fine-tuning run, obtain a stable ranking of sub-layer importance, and prune once, with no post-pruning retraining required.
> >
> > Running a new optimization after every removed layer would require L sequential fine-tuning runs for L layers, which is computationally expensive for large transformer models.
> >
> > During training, gates remain open to preserve model accuracy. The L1 regularizer does not force gates directly to zero; rather, it reduces their magnitude and increases their flip probability under the Two-Sigmoid relaxation. This mechanism reveals a ranking of sub-layer importance, making additional retraining stages unnecessary.
> > - - -
> >
> > [R1]: Quétu, Victor, et al. "LaCoOT: Layer collapse through optimal transport." Proceedings of the IEEE/CVF International Conference on Computer Vision. 2025.
> >
> > [R2]: Wen, Wei, et al. "Learning structured sparsity in deep neural networks." Advances in neural information processing systems 29 (2016).
> >
> > [R3]: Gale, Trevor, Erich Elsen, and Sara Hooker. "The state of sparsity in deep neural networks." arXiv preprint arXiv:1902.09574 (2019).
> >
> > [R4]: Mehmeti-Göpel, Christian HX Ali, and Jan Disselhoff. "Nonlinear advantage: trained networks might not be as complex as you think." International Conference on Machine Learning. PMLR, 2023.

---

### Official Review · Reviewer_3kWs · 2025-11-01

**Soundness:** 3
**Presentation:** 2
**Contribution:** 2
**Rating:** 4
**Confidence:** 5

**Summary:**

The paper proposes Structured Transformer Circuits Pruning (STCP), a single-pass pruning method that inserts learnable binary gates into each MHA and MLP sub-layer of a pretrained transformer. During fine-tuning, Gaussian noise and an L1 regularization encourage some gates to turn off, allowing the model to identify unimportant sub-layers. The method is evaluated on both vision and language transformers (CLIP, ViT-H14, BERT, RoBERTa) and reports competitive results compared with Shortened-Taylor and Joint Layer Drop.

**Strengths:**

1. The paper is well-written and easy to follow. All training details are included, and the experimental setup is consistent across datasets.
2. The experiments span both image and text domains and include ablation studies, FLOP/memory/time tables, and comparisons to multiple baselines.
3. STCP achieves competitive or slightly better results than dense baselines in several small-scale benchmarks, which is pretty impressive.

**Weaknesses:**

1. The proposed gating+L1 framework and mask-training approach have been well-studied before (e.g., SNIP, GraSP, MaskLLM, LLM-Eraser). The contribution on this side feels incremental and more engineer-like.
2. Because each sub-layer requires storing gates, masks, and gradients throughout training, the memory cost can easily exceed that of normal fine-tuning. The paper shies away from this overhead and never reports peak GPU memory. For large-scale models, such methods quickly become infeasible.
3. The pruning criterion relies on validation accuracy to decide the pruning threshold, implying that the whole fine-tuning and ranking process must be repeated for each new task, which is far from the claimed "model-agnostic" in practice.
4. All results are on CIFAR-10, Tiny-ImageNet, SST-2, and QNLI with medium-size transformers. There is no evidence that STCP works for modern-scale LLMs or vision foundation models.
5. While FLOPs and latency are reported for the inference stage, memory usage and training cost vs. accuracy trade-offs are not included, which are often the true bottlenecks for structured pruning.

**Questions:**

1. Could the authors provide the training cost, including Memory usage, Training Time, and FLOPs for baseline methods and STCP? If this is too time-consuming, at least I want to see the comparison between STCP and the dense model, i.e., regular SFT.
2. Could the authors show that their approach is really "model-agnostic"? Modern Structured pruning methods like LLM-Pruner evaluate their pruned model on several tasks instead of one.
3. Could the authors try to perform STCP on a larger-scale model like Llama3-1B/Qwen2.5-0.5B? They are not so large and are on the same model size compared to what the authors have used in their paper (ViT-H14/Roberta, etc.)

I would be happy to raise my scores if the author could provide these details.

---

> ### Author Response · Authors · 2025-11-19
>
> **[W1 - Method novelty]** Our method cannot be regarded as a simple combination of the known techniques. If only L1 regularization (group lasso) is used, the model cannot truly simulate the removal of some modules during training. This will largely affect the performance of the model after pruning. Without the differentiable Gaussian noise, binary gating would cause significant fluctuations in model performance during training, making it harder for the model to learn how to maintain performance when some modules are removed. As shown in Section 4.4, Table 3, the model’s performance can be largely improved with Gaussian noise injection.
>
> Working with both L1 regularization, Gaussian noise, and binary gating, STCP effectively learns which layers can be removed. Previous methods did not integrate these three components together and rarely achieved full-layer removal.
> - - -
> **[W2 & W5 & Q1 - Training cost]** In Table R2 (also integrated into Table 15 of the revised paper), we compare the training costs of vanilla training and STCP. During the rebuttal period, we also conducted experiments on another SOTA, LaCoOT [R1] (results updated in Table 1 of the revised paper), and have recorded and compared its training costs as well. The results show that STCP increases memory and time requirements during training compared to vanilla training. However, since the number of parameters added by the gate is very small relative to the model's overall parameter amount, the increase in requirements is minimal. And STCP continuously requires less training cost than LaCoOT.
>
> Since we employ a single-pass framework, different modules are progressively removed based on gate values without requiring post-pruning training. Therefore, in STCP, the training cost remains consistent for the same task across varying accuracy levels and numbers of layers removed.
>
> *Table R2: The peak memory usage during training and training time for image classification setups.*
> | Dataset        | Approach         | CLIP Peak Memory (GB) | CLIP Time | ViT-H14 Peak Memory (GB) | ViT-H14 Time |
> |----------------|------------------|------------------------|-----------|---------------------------|---------------|
> | CIFAR-10       | Vanilla training | 22.7                   | 3h52      | 32.64                    | 11h48        |
> | CIFAR-10       | LaCoOT           | 27.47                  | 4h26      | 46.83                    | 13h30        |
> | CIFAR-10       | STCP             | 26.64                  | 4h09      | 36.96                    | 11h07        |
> | Tiny-ImageNet  | Vanilla training | 22.59                  | 3h39      | 32.64                    | 6h52         |
> | Tiny-ImageNet  | LaCoOT           | 27.54                  | 3h53      | 46.64                    | 10h11        |
> | Tiny-ImageNet  | STCP             | 27.97                  | 3h18      | 36.9                     | 7h47         |
> | VLCS           | Vanilla training | 4.67                   | 2h10      | 20.12                    | 6h20         |
> | VLCS           | LaCoOT           | 6.93                   | 2h41      | 30.68                    | 6h31         |
> | VLCS           | STCP             | 5.39                   | 2h19      | 22.04                    | 6h13         |
> | DTD            | Vanilla training | 8.28                   | 2h40      | 32.64                    | 10h39        |
> | DTD            | LaCoOT           | 9.88                   | 2h56      | 25.21                    | 7h44         |
> | DTD            | STCP             | 10.42                  | 3h19      | 36.96                    | 14h47        |
> | Flowers 102    | Vanilla training | 7.25                   | 5h51      | 32.64                    | 21h09        |
> | Flowers 102    | LaCoOT           | 9.94                   | 6h03      | 46.6                     | 25h35        |
> | Flowers 102    | STCP             | 9.07                   | 6h00      | 36.97                    | 21h47        |
> | Aircraft       | Vanilla training | 7.48                   | 1h57      | 32.64                    | 13h12        |
> | Aircraft       | LaCoOT           | 6.92                   | 2h47      | 38.51                    | 15h48        |
> | Aircraft       | STCP             | 8.86                   | 2h34      | 36.85                    | 15h14        |
> | ImageNet       | Vanilla training | 22.71                  | 4h47      | 32.7                     | 12h56        |
> | ImageNet       | LaCoOT           | 27.67                  | 5h03      | 37.52                    | 13h58        |
> | ImageNet       | STCP             | 26.64                  | 4h18      | 37.05                    | 12h49        |

---

> > ### Author Response · Authors · 2025-11-19
> >
> > **[W4 & Q3 - Applicability on modern-scale LLMs]** Thank you for your suggestion. We trained Qwen2.5-0.5B on WikiText2 with STCP. As shown in Table R3, STCP can also effectively remove layers while maintaining model performance on modern-scale LLMs. We have also included this result in Section 4.3 of the revised version.
> >
> > *Table R3: Test performance (PPL) and the number of removed layers (Rem.) for Qwen2.5-0.5B trained on WikiText2.*
> > | Approach | PPL   | Rem.  |
> > |----------|--------|--------|
> > | Dense    | 15.09  | 0/48   |
> > | STCP     | 15.43  | 4/48   |
> > | STCP     | 17.35  | 7/48   |
> > | STCP     | 24.38  | 10/48  |
> > | STCP     | 36.27  | 13/48  |
> > | STCP     | 37.43  | 15/48  |
> > - - -
> >
> > **[W3 & Q2 - Model-agnostic]**
> > In our paper, “model-agnostic” means that the proposed pruning mechanism can be applied to different transformer models and evaluated on different downstream tasks. To address the reviewer’s concern more directly, we additionally evaluated the same pruned model (Qwen2.5-0.5B pruned with STCP on WikiText-2) on multiple tasks, including PTB, PIQA, and ARC-easy. As shown in Table R4, the pruned model maintains performance across these diverse benchmarks. This demonstrates that our approach also exhibits the kind of model-agnostic behavior highlighted by modern structured pruning methods.
> >
> > *Table R4: Evaluation of the same STCP pruned Qwen2.5-0.5B model (trained on WikiText-2) across several downstream tasks. PTB is reported in perplexity(PPL), while PIQA and ARC-easy are reported in accuracy.*
> > | Rem.  | PTB       | PIQA      | ARC_e    |
> > |-------|-----------|-----------|-----------|
> > | 0/48  | 79.17776  | 61.91513  | 49.64912  |
> > | 3/48  | 79.17776  | 61.91513  | 49.64912  |
> > | 5/48  | 90.17188  | 61.47987  | 48.77193  |
> > | 7/48  | 96.90105  | 60.06529  | 44.91228  |
> > | 9/48  | 130.6296  | 57.45375  | 38.42105  |
> > | 12/48 | 198.3316  | 56.31121  | 37.01754  |
> > | 15/48 | 461.9001  | 54.46137  | 32.45614  |
> > - - -
> >
> > [R1]: Quétu, Victor, et al. "LaCoOT: Layer collapse through optimal transport." Proceedings of the IEEE/CVF International Conference on Computer Vision. 2025.

---

> ### Comment · Reviewer_3kWs · 2025-11-22
>
> Thank you for your detailed and thorough response, but I personally still do not find the claimed novelty fully convincing. However, I do think the provided experiments have successfully addressed my other concerns, especially regarding the efficiency cost. I would suggest that the authors add those experiments to their revised version. Also, the layer numbers for Qwen2.5-0.5B should be 24 instead of 48. Given all this, as promised, I would raise my score to 6. Good luck!

---

> > ### Author Response · Authors · 2025-11-23
> >
> > Thank you very much for your follow-up and for raising your score.  You are absolutely right that Qwen2.5-0.5B has 24 transformer layers. In our paper, however, the reported number 48 refers to the total number of prunable sub-layer modules. Each layer inside Qwen2.5-0.5B contains both Multi-Head Self-Attention (MHSA) sub-layer and the MLP sub-layer, and our method applies a separate gate to each. Therefore, we have 48 prunable sub-layers in total.
> > We will make this distinction clearer in the further version to avoid confusion.

---

### Official Review · Reviewer_6Svc · 2025-11-11

**Soundness:** 3
**Presentation:** 3
**Contribution:** 2
**Rating:** 2
**Confidence:** 4

**Summary:**

This paper proposes a pruning framework named "Structured Transformer Circuits Pruning" (STCP). The method aims to learn binary gates for MHSA and MLP sub-layers within Transformer blocks through a single fine-tuning pass. Its core mechanism is claimed to combine Gaussian noise injection and L1 regularization to remove redundant sub-layers without iterative retraining. The authors report that this method outperforms SOTA (Shortened-Taylor, Joint Layer Drop) on models like CLIP, ViT, and BERT.

**Strengths:**

The paper well written, the presentation of the idea is clear for most part.
The results discussions in main text and appendix seems quite multi-dimensional and comprehensive with a lot of datasets, ablation and hyperparam discussions.
There is also a limitation section which worths encouraging.

**Weaknesses:**

1. The evaluation setup seems to have unfairness issue. One of the key advantages of this paper seems to be very cheap in finetuning compared to similar structured pruning methods like LLM-Pruner and RECAP and only requires single pass. But it is confusing that experimental detail also mentioned you still require 10000 steps to train, and there is no training cost comparison with those pruning methods. So I feel the main selling point of this paper becomes questionable.

2. Although the authors put efforts to include a lot of dataset and tasks in experiment, a key problem is that there are no any SOTA pruning methods to compare with in your main results, only a few vanilla strategies like weight pruning and gradient pruning. not sure how the performances position in the latest landscape.

3. The compression rates achieved are also underwhelming for models in main results (CLIP and ViT), i.e. up to 17/64=26.56% total pruning rate, which is quite conservative compared to SOTAs like LLM-Pruner, OBC [1].


[1] Frantar, Elias, and Dan Alistarh. "Optimal brain compression: A framework for accurate post-training quantization and pruning." Advances in Neural Information Processing Systems 35 (2022): 4475-4488.

**Questions:**

1. Does the single-pass referred to pruning steps or finetuning steps? If it's pruning stage, then it kind of defeats the whole motivation claimed in this paper, because SOTA methods like LLM-pruner also only requires single-pass for pruning stage if using taylor criterion to collect gradient. If it's finetuning step, what's the 10000 training steps mentioned in Section 4.1 refers to?
2. Related works mentioned some SOTA pruning baselines like LLM-Pruner, but none were compared in the experiment results. It is concerning if this meets ICLR standard.

---

> ### Author Response · Authors · 2025-11-19
>
> **[W1 & Q1 - Clarification of single-pass]** By single-pass, we mean one gate-aware fine-tuning run that ranks gates and prunes once, with no post-pruning retraining required.  The 10k-step schedule is a standard, unified training policy we adopt across tasks (Appendix A). Unlike iterative prune-retrain methods, our pipeline trains gates+weights once.
> LLM-Pruner and RECAP are also effective model compression methods. However, as we have mentioned in *Section.2 RELATED WORKS, Structured pruning* paragraph: both LLM-Pruner and RECAP can not remove entire MHSA or MLP modules.  Moreover, in LLM-Pruner, a lightweight LoRA fine-tuning follows pruning.  When it operates without fine-tuning, the model's performance will be significantly reduced. In contrast, our method requires only a single training session before pruning and can remove entire modules.
> - - -
> **[W2 & Q2 - Comparison with SOTA]** We have already compared with two SOTA methods: Shortened-Taylor, Joint Layer Drop. Here, we further compare our approach with LaCoOT [R1], a novel method which is recently published at ICCV 2025. The results are shown in Table R1 and have been integrated into Table 1 of the revised paper. The results demonstrate that our method exhibits significant advantages over other SOTA approaches in terms of the number of layers removed while maintaining model performance.
>
> *Table R1: Test performance (Top-1) and the number of removed sub-layers (Rem.) for all image classification tasks trained by LaCoOT and STCP.*
> | Dataset        | Approach | CLIP Top-1     | CLIP Rem. | ViT-H14 Top-1    | ViT-H14 Rem. |
> |----------------|----------|----------------|-----------|------------------|---------------|
> | CIFAR-10       | LaCoOT   | 85.44          | 2/24      | 92.22            | 8/64          |
> | CIFAR-10       | STCP     | 96.72±0.51     | 4/24      | 93.82±1.48       | 16/64         |
> | Tiny-ImageNet  | LaCoOT   | 69.86          | 1/24      | 72.64            | 8/64          |
> | Tiny-ImageNet  | STCP     | 77.67±5.32     | 3/24      | 74.48±0.07       | 11/64         |
> | VLCS           | LaCoOT   | 67.19          | 1/24      | 76.05            | 11/64         |
> | VLCS           | STCP     | 88.00±0.64     | 2/24      | 78.29±0.95       | 18/64         |
> | DTD            | LaCoOT   | 75.69          | 1/24      | 68.35            | 5/64          |
> | DTD            | STCP     | 78.96±0.17     | 3/24      | 70.23±0.46       | 11/64         |
> | Flowers 102    | LaCoOT   | 76.53          | 1/24      | 95.12            | 5/64          |
> | Flowers 102    | STCP     | 93.15±2.24     | 2/24      | 98.22±0.39       | 9/64          |
> | Aircraft       | LaCoOT   | 16.86          | 1/24      | 54.82            | 9/64          |
> | Aircraft       | STCP     | 60.81±0.17     | 2/24      | 55.71±0.32       | 17/64         |
> | ImageNet       | LaCoOT   | 43.15          | 1/24      | 71.91            | 2/64          |
> | ImageNet       | STCP     | 78.42±5.10     | 2/24      | 72.14±1.85       | 2/64          |
> - - -
> **[W3 - Maximize sparsity]** We agree that our pruning ratio is smaller than the sparsity reported for methods such as LLM-Pruner and OBC. However, these approaches prune at a much finer granularity (weights, channels, heads) inside each MHSA/MLP sub-layer and always keep the full transformer depth, whereas our method removes entire MHSA or MLP sub-layers, so the resulting network has strictly shorter forward paths.
>
> Prior works [R2, R3] on structured pruning show that coarse sparsity is more hardware-friendly than fine-grained sparsity. In this sense, our method is complementary to fine-grained approaches like LLM-Pruner and OBC: they maximise nominal sparsity at small granularity, while we push sparsity at the level of whole sub-layers in a way that is directly useful for deployment-time acceleration.
> - - -
> [R1]: Quétu, Victor, et al. "LaCoOT: Layer collapse through optimal transport." Proceedings of the IEEE/CVF International Conference on Computer Vision. 2025.
>
> [R2]: Wen, Wei, et al. "Learning structured sparsity in deep neural networks." Advances in neural information processing systems 29 (2016).
>
> [R3]: Gale, Trevor, Erich Elsen, and Sara Hooker. "The state of sparsity in deep neural networks." arXiv preprint arXiv:1902.09574 (2019).

---

### Meta-Review · Area_Chair_6yqB · 2026-01-06

**Summary:**

The reviewers raised significant concerns regarding the novelty, efficiency, and comparative landscape of the proposed Structured Transformer Circuits Pruning (STCP) framework. While the method introduces binary gates with Gaussian noise and $L_{1}$ regularization to prune MHSA and MLP sub-layers in a single pass, Reviewer 3kWs and Reviewer 6Svc noted that these components are well-studied in literature (e.g., SNIP, MaskLLM), rendering the contribution incremental. A primary point of contention was the "single-pass" claim; reviewers pointed out that a 10,000-step fine-tuning schedule is substantial, potentially negating the perceived "cheapness" of the pruning process. Furthermore, the achieved compression rates for models like CLIP and ViT (approximately 26.6%) were viewed as underwhelming compared to contemporary state-of-the-art (SOTA) methods like LLM-Pruner or OBC, which achieve higher sparsity while maintaining similar performance. Despite the authors' arguments regarding hardware-friendly coarse sparsity, the overall performance-to-compression trade-off remained a critical weakness that informed the final decision.

**Reviewer Concerns:**

The rebuttal phase addressed several empirical gaps, though fundamental conceptual concerns remain outstanding.

## **Addressed Concerns**:

**Training Cost and Memory**: The authors provided detailed comparisons of peak GPU memory and training time against vanilla training and the LaCoOT baseline. The results show that STCP has minimal memory overhead (e.g., 26.64 GB vs. 22.7 GB for CLIP on CIFAR-10) and is generally more efficient than LaCoOT.


## **Outstanding Concerns**:

**Novelty**: Even the reviewer who upgraded their score (3kWs) explicitly stated that the claimed novelty was not fully convincing, as the framework relies on standard gating and regularization techniques.

**Scalability to Modern LLMs**: Although the authors validate the proposed method to Qwen2.5-0.5B during the rebuttal phase, the performance on modern-scale LLM is still largely unclear in the current status.

**Competitive Landscape**: Reviewer TxxT’s concern regarding structured pruning that removes neurons/width instead of entire layers remains a point of philosophical disagreement; the authors did not provide a direct head-to-head comparison showing STCP's accuracy advantage over these more flexible structured methods.

**Reviewer Scores:**

Based on the full discussion and rebuttal materials:

Reviewer **3kWs**: Formally increased their score from 4 to 6 (Accept), acknowledging the successful empirical response regarding efficiency and scaling, despite lingering novelty concerns.

Reviewer **6Svc**: Originally a 2 (Reject). While the authors clarified the single-pass definition and provided SOTA comparisons with LaCoOT, the core critique of underwhelming compression rates and the 10k training steps likely would have limited a score increase.

Reviewer **TxxT**: Originally a 2 (Reject). The addition of quantization and modern LLM experiments addressed their breadth concerns. However, their fundamental interest in comparisons with structured width-pruning (like LLM Surgeon) was not fully met, likely resulting in a final estimation of 3.

While the authors were professional and diligent in providing new experiments (specifically for LLMs and quantization), the paper's core methodology does not represent a sufficient leap in novelty or efficiency to meet the high bar of ICLR. The "single-pass" advantage is mitigated by the standard fine-tuning length required, and the pruning effectiveness at high ratios is notably lower than existing fine-grained or structured-width alternatives

---

### Decision · Program_Chairs · 2026-01-26

Reject